# CamSAM2: Segment Anything Accurately in Camouflaged Videos

Yuli Zhou[1,3]    Yawei Li[1]    Yuqian Fu[4]    Luca Benini[1,5]    Ender Konukoglu[1]    Guolei Sun[2*]

[1]ETH Zurich    [2]Nankai University    [3]University of Zurich
[4]INSAIT, Sofia University "St. Kliment Ohridski"    [5]University of Bologna

## Abstract

Video camouflaged object segmentation (VCOS), aiming at segmenting camouflaged objects that seamlessly blend into their environment, is a fundamental vision task with various real-world applications. With the release of SAM2, video segmentation has witnessed significant progress. However, SAM2's capability of segmenting camouflaged videos is suboptimal, especially when given simple prompts such as point and box. To address the problem, we propose **Cam**ouflaged **SAM2** (**CamSAM2**), which enhances SAM2's ability to handle camouflaged scenes without modifying SAM2's parameters. Specifically, we introduce a decamouflaged token to provide the flexibility of feature adjustment for VCOS. To make full use of fine-grained and high-resolution features from the current frame and previous frames, we propose implicit object-aware fusion (IOF) and explicit object-aware fusion (EOF) modules, respectively. Object prototype generation (OPG) is introduced to abstract and memorize object prototypes with informative details using high-quality features from previous frames. Extensive experiments are conducted to validate the effectiveness of our approach. While CamSAM2 only adds negligible learnable parameters to SAM2, it substantially outperforms SAM2 on three VCOS datasets, especially achieving *12.2* mDice gains with click prompt on MoCA-Mask and *19.6* mDice gains with mask prompt on SUN-SEG-Hard, with Hiera-T as the backbone. The code is available at https://github.com/zhoustan/CamSAM2.

## 1 Introduction

Camouflaged object detection (COD) and video camouflaged object segmentation (VCOS) aim to identify objects that blend seamlessly into their surroundings. Unlike standard object segmentation tasks, where objects typically exhibit clear boundaries and contrast with the background, camouflaged objects are naturally indistinguishable from the background. These tasks have various applications in wildlife monitoring, surveillance, and search-and-rescue operations [1, 2]. COD focuses on detecting camouflaged objects in individual images, while VCOS extends it to video sequences, adding the complexity of modeling temporal information across frames. Despite recent advancements in COD [3, 4, 5, 6, 7, 8, 9] and VCOS [3, 4, 10, 11, 12, 13, 14], the performance remains far from satisfactory compared to general segmentation tasks.

The recently introduced vision foundation model, Segment Anything Model 2 (SAM2) [15], marks a significant advancement in video segmentation. SAM2 has learned rich and generalizable representations for natural scenes on the SA-1B [16] (11M images, 1B masks) and SA-V [15] (50.9K videos, 35.5M masks) datasets. Therefore, its features are optimized for natural scenes, while SAM2's ability of segmenting camouflaged objects is suboptimal, as in [17, 18]. Following the setting of SAM2, we apply point, box, or mask prompts on the first frame of each video. Specifically, we randomly

---

*Corresponding author.

39th Conference on Neural Information Processing Systems (NeurIPS 2025).

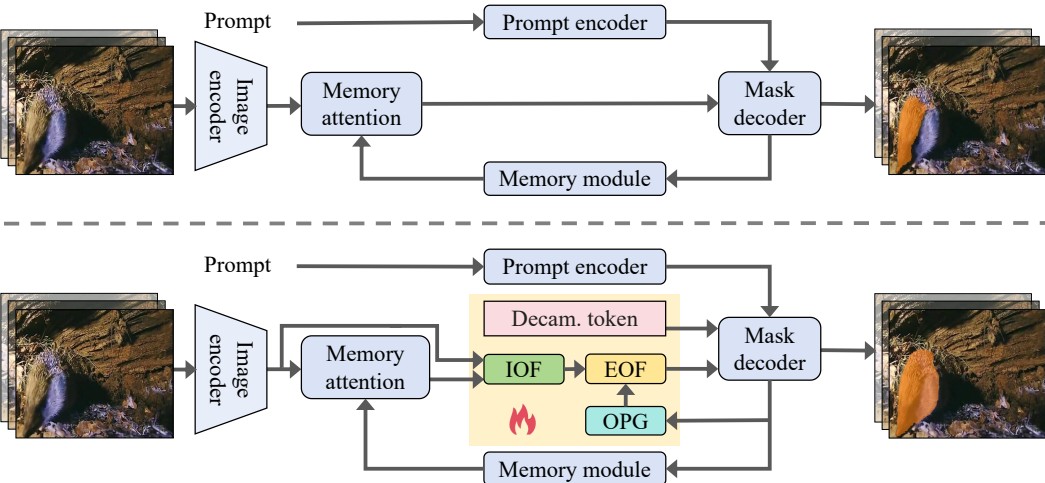

Figure 1: **Illustration of SAM2 and CamSAM2**. *Top:* SAM2's segmentation of the camouflaged object is suboptimal, primarily because its feature optimization is biased toward natural videos, and its design does not account for the unique challenges inherent to VCOS. *Bottom:* CamSAM2 improves SAM2's ability to segment and track camouflaged objects by introducing a *decamouflaged token*, *IOF* to enhance features with high-resolution features, and *EOF* and *OPG* to further enhance features by exploiting informative object details across time. CamSAM2 only adds a limited number of parameters to SAM2 while keeping all SAM2's parameters fixed and fully inheriting SAM2's zero-shot ability. The segmentation result is overlaid in orange on the frame.

sample a point within the ground truth region as a point prompt, extract the tight bounding box as the box prompt, or use the full ground-truth mask as the mask prompt. As shown in Fig. 1, with a point prompt, SAM2 segments only part of a camouflaged animal (*hedgehog*), indicating that there is still room for improvements in VCOS.

This paper aims to develop a model for accurate segmentation in camouflaged videos, requiring both natural image understanding and effective identification of camouflaged objects in complex environments. To achieve this, we identify two core challenges in adapting SAM2 for VCOS: **(1)** SAM2 is optimized for natural scenes rather than camouflaged environments. **(2)** The architecture does not account for the complexities of segmenting and tracking camouflaged objects across time. For VCOS, accurately segmenting camouflaged objects for a frame requires: *a)* exploiting fine-grained and detailed features from the frame, and *b)* considering the temporal evolvement of fine-grained features from previous frames. For exploiting temporal information, SAM2 is equipped with a memory module containing a memory encoder and a memory bank. However, only low-resolution and coarse features are encoded into the memory, which is suboptimal for accurate VCOS.

To tackle the above limitations and fully keep SAM2's ability to process natural videos, we introduce **Cam**ouflaged **SAM2**, dubbed as **CamSAM2**, equipping SAM2 with the ability to effectively tackle VCOS, as depicted in Fig. 1. CamSAM2 includes a learnable *decamouflaged token*, which extends the token structure of SAM2 and provides flexibility to optimize features for VCOS without modifying SAM2's trained parameters. To exploit the fine-grained features of the frame, we propose the *Implicit Object-aware Fusion* (IOF) module, which leverages high-resolution features from the early layers of the image encoder to enhance the model's perception of fine-grained details. To make use of detailed features from previous frames, we further propose *Object Prototype Generation* (OPG) to abstract high-quality features within the object region into informative object prototypes through Farthest Point Sampling (FPS) and *k*-means. Those object prototypes are saved to memory for easy usage by the *Explicit Object-aware Fusion* (EOF) module that is designed to integrate explicit object-aware information across the temporal dimension. Our design avoids saving the high-resolution features in the memory and only adds negligible computations to SAM2 while accounting for a large amount of temporal information.

We conduct extensive experiments in §4 to validate the effectiveness of CamSAM2 on three VCOS benchmarks: two camouflaged animal datasets, MoCA-Mask [4] and CAD [19], and one camouflaged medical dataset, SUN-SEG [20]. Our experiments show that CamSAM2 significantly outperforms SAM2 by achieving improvements of *12.2/13.1* mDice scores with click prompt on MoCA-Mask for

Hiera-T/Hiera-S backbones, and *19.6* mDice gains with mask prompt on SUN-SEG-Hard for Hiera-T backbone. When directly evaluating CamSAM2 on CAD without further finetuning, we observe strong zero-shot ability. Since all SAM2's weights remain unchanged, CamSAM2 totally inherits SAM2's capability on segmenting natural videos. In summary, our contributions are three-fold:

(1) We propose CamSAM2 to equip SAM2 with the ability to segment and track camouflaged objects in videos while keeping SAM2's strong generalizability in natural videos. (2) CamSAM2 introduces a decamouflaged token to achieve easy feature adjustments for the VCOS task without affecting SAM2's trained weights. To effectively exploit the crucial fine-grained and high-resolution features from both the current frame and previous frames, we propose IOF, EOF, and OPG modules. (3) Our approach clearly outperforms SAM2 and sets new state-of-the-art performance on public VCOS datasets. Experiments also show the strong zero-shot ability of CamSAM2 in the domain of VCOS.

## 2 Related work

### 2.1 Camouflaged object detection

Camouflaged Scene Understanding (CSU) focuses on interpreting scenes where objects blend closely with their backgrounds, such as natural environments like forests, oceans, and deserts. Early works in this field primarily involved the collection of extensive image and video datasets, such as CAMO [21], COD10K [22], NC4K [23], CAD [19], MoCA-Mask [4], and MoCA-Mask-Pseudo [4], which laid the foundation for CSU. Traditional COD methods extract foreground-background features using optical features [24], color, and texture [25]. Deep learning has advanced COD with CNNs and transformers. SINet [22] and SINet-V2 [26] enhance fine-grained cues by applying receptive fields and texture-enhanced modules, while DQNet [27] applies cross-modal detail querying to detect subtle features. Transformer-based models like CamoFormer [9] leverage multi-scale feature extraction with masked separable attention, and WSSCOD [5] employs a frequency transformer and noisy pseudo labels for weak supervision. ZoomNeXt [3] further optimizes multi-scale extraction via a collaborative pyramid network. These advancements refine COD by integrating sophisticated architectures and diverse supervision strategies.

### 2.2 Video camouflaged object segmentation

VCOS [14, 28, 12, 29] extends COD to videos, introducing challenges from motion, dynamic backgrounds, and temporal consistency. Former VCOS models tackle these with motion learning, spatial-temporal attention, and advanced segmentation techniques to maintain object coherence across frames. Motion-guided models enhance segmentation by leveraging motion cues. IMEX [30] integrates implicit and explicit motion learning for robust detection. TMNet [28] refines motion features with a transformer-based encoder and neighbor connection decoder. Flow-SAM [31] uses optical flow as input or a prompt, guiding SAM to detect moving camouflaged objects. Spatial-temporal attention enhances the tracking of camouflaged objects. TSP-SAM [10] and SAM-PM [11] improve SAM's ability to detect subtle movements. Static-Dynamic-Interpretability [12] quantifies static and dynamic information in spatial-temporal models, aiding balanced approaches. Assessing camouflage quality is also essential for VCOS. CAMEVAL [32] introduces scores evaluating background similarity and boundary visibility, refining datasets and improving model robustness. These advancements drive more accurate and effective VCOS systems.

### 2.3 Segment Anything Model 2

SAM2 [15] is a vision foundation model for promptable segmentation across images and videos. Compared to SAM [16], which is limited to image segmentation, SAM2 offers a significant performance leap in video segmentation SAM2 has demonstrated strong capabilities in many tasks, including medical image, video and 3D segmentation [33, 34, 35, 36, 37, 38, 39, 40], video object tracking and segmentation [41, 42], remote sensing [43], 3D mesh and point cloud segmentation [44], COD and VCOS [17, 38, 39, 18]. In previous works [17, 18], although SAM2's performance on camouflaged video segmentation has surpassed most existing methods, these studies primarily focused on direct evaluation, lightweight fine-tuning, or integrating SAM2 with multimodal large language models. However, they did not address the fundamental architectural challenges of adapting SAM2 to VCOS,

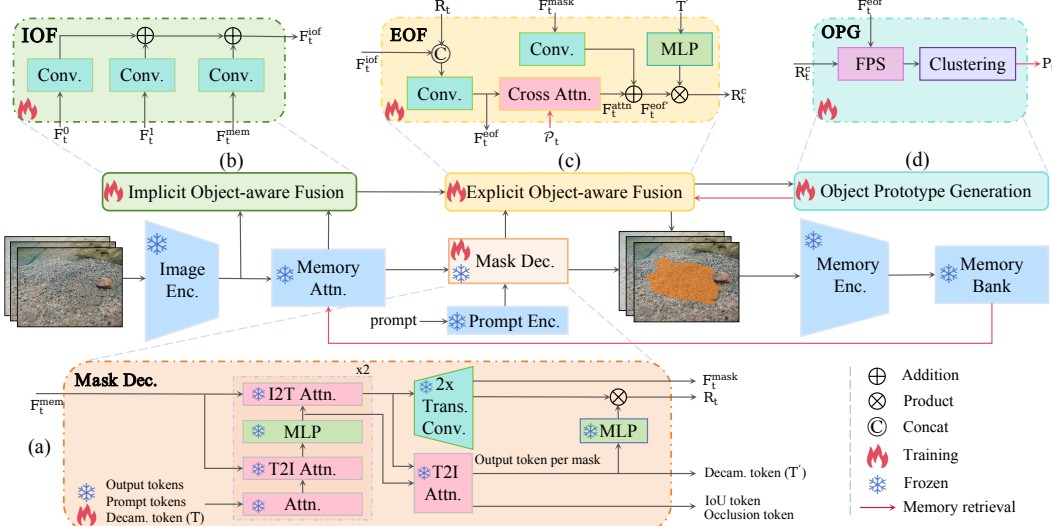

Figure 2: **Overall architecture of CamSAM2.** CamSAM2 effectively captures and segments camouflaged objects by leveraging implicit and explicit object-aware information from the current or previous frames. It includes the following key components: *(a)* the *decamouflaged token*, which extends SAM2's token structure to learn features suitable for camouflaged objects; *(b)* an *IOF* module to enrich memory-conditioned features with implicitly object-aware high-resolution features; *(c)* an *EOF* module to aggregate explicit object-aware features; and *(d)* an *OPG* module, generating informative object prototypes, which guides cross-attention in EOF. These components work together to preserve fine details, enhance segmentation quality, and track camouflaged objects across time.

leaving a significant performance gap compared to other VOS tasks, especially when using simple prompts.

## 3 Method

We propose CamSAM2, equipping SAM2 with the ability to accurately segment camouflaged objects in videos while retaining SAM2's original capabilities. §3.1 briefly reviews the architecture of SAM2. From §3.2 to §3.5, we describe CamSAM2 tailored for VCOS. With fixing SAM2's parameters, CamSAM2 proposes a learnable decamouflaged token, Implicit and Explicit Object-aware Fusion, and Object Prototype Generation to enhance feature representations, thus leading to improved performance, as shown in Fig. 2.

### 3.1 Preliminaries

SAM2 [15] is a pioneering vision foundation model designed for promptable visual segmentation tasks. Different from SAM [16], SAM2 includes a memory module that stores information about the object from previous frames. It contains an image encoder, memory attention, prompt encoder, mask decoder, memory encoder, and memory bank. For each frame, the image encoder extracts representative visual features, which are then conditioned on the features and predictions of past frames. If a point or box prompt is given, the prompt encoder encodes it into sparse or dense embeddings, then segments the prompted frame; if a mask prompt is given, SAM2 directly uses the mask as the current frame output. Exploiting memory-conditioned features and prompt embeddings, the mask decoder outputs the segmentation mask. The memory encoder then updates the memory bank with the output mask and the unconditioned frame embedding to support the segmentation of subsequent frames. SAM2 is pre-trained on SA-1B [16] and further trained on SA-V [15], achieving strong performance across video and image segmentation tasks. For more details, please refer to [15].

## 3.2 Decamouflaged token

Given a video clip containing $m$ frames, we denote all frames as $\{\mathbf{I}_{t-m+1}, \cdots, \mathbf{I}_i, \cdots, \mathbf{I}_t\}$ with ground-truth segmentation masks of $\{\mathbf{S}_{t-m+1}, \cdots, \mathbf{S}_i, \cdots, \mathbf{S}_t\}$. Especially, $\mathbf{I}_t$ is the current frame for the purpose of easy explanation. We use the image encoder to extract features for all frames, denoted as $\{\mathbf{F}_{t-m+1}, \cdots, \mathbf{F}_i, \cdots, \mathbf{F}_t\}$. Here, $\mathbf{F}_i$ can be further represented as $\{\mathbf{F}_i^0, \cdots, \mathbf{F}_i^j, \cdots, \mathbf{F}_i^{L-1}\}$, containing feature maps extracted from $L$ different intermediate layers, where $\mathbf{F}_i^j \in \mathbb{R}^{c_j \times h_j \times w_j}$, with $c_j$, $h_j$, and $w_j$ representing channels, height, and width, respectively.

SAM2's output tokens include an object score (occlusion) token, an IoU token, and mask tokens. To enhance SAM2's ability of segmenting camouflaged objects, we introduce a new learnable decamouflaged token $\mathbf{T} \in \mathbb{R}^{1 \times 256}$, enabling it to optimize features for segmenting camouflaged objects. As depicted in Fig. 2, integrated with SAM2's output tokens, the decamouflaged token undergoes the same layers as output tokens within SAM2's mask decoder. After this, the output decamouflaged token is denoted as $\mathbf{T}'$. This token is updated through back-propagated gradients, while SAM2's weights remain frozen. $\mathbf{T}'$ is then passed through an MLP layer to participate in computing CamSAM2's final mask logits, which will be explained in §3.4.

## 3.3 Implicit object-aware fusion

Early-layer features from the image encoder capture *high-resolution details*, such as edges and textures, essential for distinguishing subtle differences between the camouflaged object and the background. These early-layer features are *implicit* object-aware, as features for background and non-relevant objects also exist with similar magnitude. In contrast, deeper layers focus on high-level semantic information. In SAM2, memory-conditioned features are computed by *only* conditioning high-level semantic features, without using detailed features from early layers. To this end, we propose an IOF module to fuse these implicit object-aware features with memory-conditioned features.

For SAM2, three feature maps from the Hiera image encoder [45] are extracted for each frame, *i.e.*, $L = 3$. We have $\mathbf{F}_t^0$, $\mathbf{F}_t^1$, and $\mathbf{F}_t^2$ for the current frame $\mathbf{I}_t$. We denote the memory-conditioned feature as $\mathbf{F}_t^{mem}$, encoded by the memory-attention module on $\mathbf{F}_t^2$, as in [15]. The high-resolution features $\mathbf{F}_t^0$ and $\mathbf{F}_t^1$ are fused with $\mathbf{F}_t^{mem}$ via compression modules and point-wise addition to create a refined feature representation $\mathbf{F}_t^{iof} \in \mathbb{R}^{c_0 \times h_0 \times w_0}$, where a compression module $C(\cdot)$ consists of two convolutional layers, followed by an upsampling layer. This process is given by:

$$\mathbf{F}_t^{iof} = C_0(\mathbf{F}_t^0) + C_1(\mathbf{F}_t^1) + C_2(\mathbf{F}_t^{mem}). \tag{1}$$

## 3.4 Explicit object-aware fusion

After obtaining $\mathbf{F}_t^{iof}$, we further refine it by EOF, which exploits *explicit* object-aware information from the current frame and previous frames, through employing object mask logits and object prototypes (see §3.5). We have three steps to fuse informative features. *First*, feature $\mathbf{F}_t^{iof}$, with shape $\mathbb{R}^{c_0 \times h_0 \times w_0}$, is directly concatenated with SAM2's mask logits $\mathbf{R}_t$, which has shape $\mathbb{R}^{1 \times h_0 \times w_0}$. This concatenated feature is then processed through a convolutional layer to reduce the channels back to $c_0$, resulting in the output with the original shape $\mathbb{R}^{c_0 \times h_0 \times w_0}$, denoted as:

$$\mathbf{F}_t^{eof} = \text{Conv}\left(\left[\mathbf{F}_t^{iof}; \mathbf{R}_t\right]\right). \tag{2}$$

*Second*, $\mathbf{F}_t^{eof}$ goes through a cross-attention layer. Prototypes generated from previous frames, representing clustered camouflaged features, serve as informative priors to help distinguish the camouflaged object from its background. A cross-attention mechanism takes $\mathbf{F}_t^{eof}$ as a query, and leverages these prototypes as keys and values, effectively exploiting the information within the object prototypes to refine $\mathbf{F}_t^{eof}$. This design naturally suppresses outdated or irrelevant prototypes: if a prototype is inconsistent or suboptimal (e.g., due to occlusion or missing), it receives lower attention and contributes less to the output. This cross-attention mechanism provides robustness and avoids overfitting to specific regions. Formally, we update $\mathbf{F}_t^{eof}$ by conducting cross-attention with prototypes $\mathcal{P}_t = \{\mathbf{P}_0, \mathbf{P}_1, \ldots, \mathbf{P}_{t-1}\}$ from previous frames, given by:

$$\mathbf{F}_t^{attn} = \text{Attn}(\mathbf{F}_t^{eof}, \mathcal{P}_t, \mathcal{P}_t). \tag{3}$$

*Third*, the attention-refined feature $\mathbf{F}_t^{attn}$ is combined with the upscaled mask feature $\mathbf{F}_t^{mask}$ from SAM2 mask decoder. The upscaled mask feature is first processed through a convolutional layer and then fused with $\mathbf{F}_t^{attn}$ via point-wise addition, as follows:

$$\mathbf{F}_t^{eof\prime} = \mathbf{F}_t^{attn} + \text{Conv}(\mathbf{F}_t^{mask}). \tag{4}$$

*Finally*, we calculate the mask logits $\mathbf{R}_t^c$ of CamSAM2, by processing the output decamouflaged token $\mathbf{T}'$ through an MLP layer, then performing point-wise product with the $\mathbf{F}_t^{eof\prime}$, as shown below:

$$\mathbf{R}_t^c = \text{MLP}(\mathbf{T}') \cdot \mathbf{F}_t^{eof\prime}. \tag{5}$$

This approach incorporates both implicit and explicit camouflaged information, which can enhance mask generation for more accurate segmentation for the VCOS task.

### 3.5   Object prototype generation

To effectively represent the camouflaged features within the mask (object) region, we employ Farthest Point Sampling (FPS) [46] to identify $k$ points within the predicted mask region, which act as cluster centers. This approach ensures that the sampled points are well-distributed throughout the mask, capturing diverse and important characteristics of the camouflaged object. Then, we group all pixels in the predicted mask region into $k$ clusters by conducting one-iteration $k$-means, using the sampled $k$ points as initial centers. The prototype of each cluster is represented as the mean of the spatial features of the points in the cluster. This prototype generation process is denoted as $\mathcal{F}_p$, as shown in:

$$\mathbf{P}_t = \{P_t^i \mid 1 \leq i \leq k\} = \mathcal{F}_p(\mathbf{F}_t^{eof}, \mathbf{R}_t^c), \tag{6}$$

where $\mathbf{P}_t$ represents the camouflaged object prototypes extracted from high-resolution and detailed features for the frame $\mathbf{I}_t$. The prototypes are concatenated and then saved in the memory, which will be used by EOF (§3.4) when segmenting the subsequent frames.

## 4   Experiments

### 4.1   Experimental setup

**Datasets.**   Our experiments are conducted on three video datasets: two popular camouflaged animal datasets, MoCA-Mask [4] and CAD [19], and one camouflaged medical dataset, SUN-SEG [20]. The pioneering Moving Camouflaged Animals dataset (MoCA) [47] comprises 37K frames from 141 YouTube video sequences. The dataset MoCA-Mask is reorganized from the MoCA, containing 71 video sequences with 19,313 frames for training and 16 video sequences with 3,626 frames for testing, respectively, with pixel-wise ground-truth masks on every five frames. It also generates a MoCA-Mask-Pseudo dataset, which contains pseudo masks for unlabeled frames with a bidirectional optical-flow-based consistency check strategy. The Camouflaged Animal Dataset (CAD) includes 9 short videos in total that have 181 hand-labeled masks on every five frames. SUN-SEG is the largest benchmark for video polyp segmentation, derived from SUN-database [48]. It consists of a training set with 112 clips (19,544 frames) and two test sets: SUN-SEG-Easy, containing 119 clips (17,070 frames), and SUN-SEG-Hard, comprising 54 clips (12,522 frames).

**Training and inference.**   We simulate interactive prompting of the model in the training process, prompting on the first frame of the sampled sequence. Following the training strategy of SAM2, we use three types of prompts (mask, bounding box, 1-click point of foreground) for training, with the probabilities of 0.5, 0.25, and 0.25, respectively.

To train the model, we use a combined loss of binary cross-entropy (BCE) and dice loss for mask predictions across the entire video. This loss applies to both SAM2's mask logits $\mathbf{R}_i$ and the CamSAM2's mask logits $\mathbf{R}_i^c$, compared with the ground-truth mask $\mathbf{S}_i$ of frame $\mathbf{I}_i$, as follows:

$$\mathcal{L}_C = \sum_{i=t-m+1}^{t} \left[ \mathcal{L}_{BCE}(\mathbf{R}_i, \mathbf{S}_i) + \mathcal{L}_{BCE}(\mathbf{R}_i^c, \mathbf{S}_i) \right],$$

$$\mathcal{L}_D = \sum_{i=t-m+1}^{t} \left[ \mathcal{L}_{Dice}(\mathbf{R}_i, \mathbf{S}_i) + \mathcal{L}_{Dice}(\mathbf{R}_i^c, \mathbf{S}_i) \right],$$

$$\mathcal{L} = \mathcal{L}_C + \mathcal{L}_D, \tag{7}$$

Table 1: **Comparisons between our method and existing approaches on MoCA-Mask.** CamSAM2 outperforms the existing method by achieving new state-of-the-art performance. "SAM2-FT" refers to a version of SAM2 in which the mask decoder is fine-tuned. The results of all these methods (excluding SAM2) are from the corresponding publications. The best results are shown in **bold**. ↑: the higher the better, ↓: the lower the better.

| Model | Backbone | Params (M) | Prompt | $S_m$ ↑ | $F_\beta^\omega$ ↑ | MAE ↓ | $F_\beta$ ↑ | $E_m$ ↑ | mDice ↑ | mIoU ↑ |
|---|---|---|---|---|---|---|---|---|---|---|
| **EGNet**[2019] [55] | ResNet-50 | 111.7 | - | 54.7 | 11.0 | 3.5 | 13.6 | 57.4 | 14.3 | 9.6 |
| **BASNet**[2019] [56] | ResNet-50 | 87.1 | - | 56.1 | 15.4 | 4.2 | 17.3 | 59.8 | 19.0 | 13.7 |
| **CPD**[2019] [57] | ResNet-50 | 47.9 | - | 56.1 | 12.1 | 4.1 | 15.2 | 61.3 | 16.2 | 11.3 |
| **PraNet**[2020] [58] | ResNet-50 | 32.6 | - | 61.4 | 26.6 | 3.0 | 29.6 | 67.4 | 31.1 | 23.4 |
| **SINet**[2020] [22] | ResNet-50 | 48.9 | - | 59.8 | 23.1 | 2.8 | 25.6 | 69.9 | 27.7 | 20.2 |
| **SINet-V2**[2021] [26] | Res2Net-50 | 27.0 | - | 58.8 | 20.4 | 3.1 | 22.9 | 64.2 | 24.5 | 18.0 |
| **PNS-Net**[2021] [59] | ResNet-50 | 142.9 | - | 52.6 | 5.9 | 3.5 | 8.4 | 53.0 | 8.4 | 5.4 |
| **RCRNet**[2019] [60] | ResNet-50 | 53.8 | - | 55.5 | 13.8 | 3.3 | 15.9 | 52.7 | 17.1 | 11.6 |
| **MG**[2021] [61] | VGG | 4.8 | - | 53.0 | 16.8 | 6.7 | 19.5 | 56.1 | 18.1 | 12.7 |
| **SLT-Net-LT**[2022] [4] | PVTv2-B5 | 82.3 | - | 63.1 | 31.1 | 2.7 | 33.1 | 75.9 | 36.0 | 27.2 |
| **ZoomNeXt**[2024] [3] | PVTv2-B5 | 84.8 | - | 73.4 | 47.6 | 1.0 | 49.7 | 73.6 | 49.7 | 42.2 |
| **SAM2**[2024] [15] | Hiera-T | 38.9 | 1-click | 68.2 | 50.7 | 7.7 | 52.5 | 73.6 | 52.1 | 44.8 |
| **SAM2-FT** | Hiera-T | 38.9 | 1-click | 70.2 | 54.5 | 7.2 | 56.0 | 76.8 | 55.2 | 48.1 |
| **CamSAM2** | Hiera-T | 39.4 | 1-click | 75.2 | 61.7 | 7.3 | 63.7 | 82.0 | 64.3 | 54.6 |
| **SAM2**[2024] [15] | Hiera-T | 38.9 | box | 81.5 | 69.9 | 0.6 | 70.9 | 89.4 | 72.7 | 62.3 |
| **SAM2-FT** | Hiera-T | 38.9 | box | 82.1 | 71.6 | 0.5 | 72.8 | 90.8 | 73.6 | 63.4 |
| **CamSAM2** | Hiera-T | 39.4 | box | 82.9 | 72.4 | 0.6 | 73.2 | 94.2 | 75.5 | 64.8 |
| **SAM-PM**[2024] [11] | ViT-L | 303.0 | mask | 72.8 | 56.7 | 0.9 | - | 81.3 | 59.4 | 50.2 |
| **SAM2**[2024] [15] | Hiera-T | 38.9 | mask | 84.7 | 76.0 | **0.4** | 76.9 | 91.9 | 77.1 | 67.9 |
| **SAM2-FT** | Hiera-T | 38.9 | mask | 84.7 | 76.2 | **0.4** | 77.1 | 91.9 | 77.2 | 68.0 |
| **CamSAM2** | Hiera-T | 39.4 | mask | **86.2** | **78.7** | **0.4** | **79.6** | **96.2** | **80.2** | **70.5** |

Table 2: **Detailed comparisons between SAM2 and CamSAM2 on MoCA-Mask.** CamSAM2 *consistently* outperforms SAM2 for all considered prompt types and backbones. Improvements of CamSAM2 over SAM2 are shown in dark green.

| Model | Prompt | Hiera-T | | Hiera-S | |
|---|---|---|---|---|---|
| | | mDice ↑ | mIoU ↑ | mDice ↑ | mIoU ↑ |
| **SAM2** | 1-click | 52.1 | 44.8 | 54.9 | 46.7 |
| **CamSAM2** | | **64.3** (+12.2) | **54.6** (+9.8) | **68.0** (+13.1) | **58.8** (+12.1) |
| **SAM2** | box | 72.7 | 62.3 | 73.7 | 63.8 |
| **CamSAM2** | | **75.5** (+2.8) | **64.8** (+2.5) | **76.4** (+2.7) | **66.1** (+2.3) |
| **SAM2** | mask | 77.1 | 67.9 | 80.3 | 70.7 |
| **CamSAM2** | | **80.2** (+3.1) | **70.5** (+2.6) | **81.4** (+1.1) | **71.7** (+1.0) |

where $\mathcal{L}$ is the final loss for our approach, summing the BCE loss $\mathcal{L}_C$ and the dice loss $\mathcal{L}_D$.

During inference, we provide a prompt at the first frame of a video, following [15, 11]. Our final output is the average of the logits of SAM2 and CamSAM2 masks for the error correction. For more training and inferencing details, please see Appendix A.5.

**Implementation details.** The proposed CamSAM2 is implemented with PyTorch [49]. CamSAM2 is initialized with the parameters of SAM2. We freeze all parameters used in SAM2 and initialize other parameters randomly. We set $betas = (0.9, 0.999)$ for the optimizer Adam and use the learning rate of 1e-3. We train CamSAM2 on 4 NVIDIA RTX 4090 GPUs for 10 epochs. For camouflaged animal segmentation, we train the model using the MoCA-Mask-Pseudo training set and evaluate it on the MoCA-Mask test set and CAD. During inference, we apply the 1-click, box, and mask prompts only on the first frame of each video. For camouflaged polyp segmentation, we train the model using the SUN-SEG training set and perform inference using the mask prompt on the first frame of each video on the SUN-SEG-Easy and SUN-SEG-Hard test sets.

**Evaluation metrics.** We adopt seven evaluation metrics to measure the quality of predicted pixel-wise masks: S-measure ($S_m$) [50], F-measure ($F_\beta$) [51], weighted F-measure ($F_\beta^\omega$) [52], mean absolute error (MAE) [53], E-measure ($E_m$) [54], mean Dice (mDice), and mean IoU (mIoU).

## 4.2 Experimental results

**Results on MoCA-Mask.** Tab. 1 compares four promptable methods on MoCA-Mask. For a fair comparison, we fine-tune the mask decoder (4.2M parameters) of SAM2 on MoCA-Mask. CamSAM2

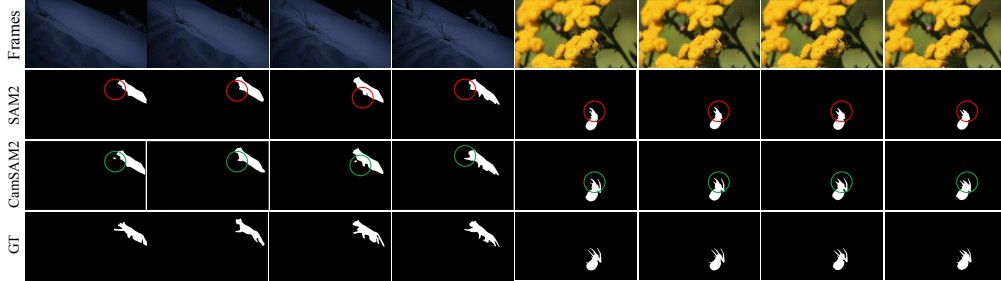

Figure 3: **Qualitative comparisons between SAM2 and CamSAM2 using 1-click prompt with the Hiera-T backbone on two MoCA-Mask clips.** From *top* to *bottom*: the input frames, SAM2's results, CamSAM2's results, and ground-truth masks. CamSAM2 demonstrates improved accuracy in VCOS, especially in complex backgrounds, as shown by the circles. *Best viewed in color.*

Table 3: **Comparisons between our method and existing approaches on CAD.**

| Model | Backbone | Params (M) | Prompt | $S_m \uparrow$ | $F_\beta^\omega \uparrow$ | MAE $\downarrow$ | $F_\beta \uparrow$ | $E_m \uparrow$ | mDice $\uparrow$ | mIoU $\uparrow$ |
|---|---|---|---|---|---|---|---|---|---|---|
| **EGNet**[2019] [55] | ResNet-50 | 111.7 | - | 61.9 | 29.8 | 4.4 | 35.0 | 66.6 | 32.4 | 24.3 |
| **BASNet**[2019] [56] | ResNet-50 | 87.1 | - | 63.9 | 34.9 | 5.4 | 39.4 | 77.3 | 39.3 | 29.3 |
| **CPD**[2019] [57] | ResNet-50 | 47.9 | - | 62.2 | 28.9 | 4.9 | 35.7 | 66.7 | 33.0 | 23.9 |
| **PraNet**[2020] [58] | ResNet-50 | 32.6 | - | 62.9 | 35.2 | 4.2 | 39.7 | 76.3 | 37.8 | 29.0 |
| **SINet**[2020] [22] | ResNet-50 | 48.9 | - | 63.6 | 34.6 | 4.1 | 39.5 | 77.5 | 38.1 | 28.3 |
| **SINet-V2**[2021] [26] | Res2Net-50 | 27.0 | - | 65.3 | 38.2 | 3.9 | 43.2 | 76.2 | 41.3 | 31.8 |
| **PNS-Net**[2021] [59] | ResNet-50 | 142.9 | - | 65.5 | 32.5 | 4.8 | 41.7 | 67.3 | 38.4 | 29.0 |
| **RCRNet**[2019] [60] | ResNet-50 | 53.8 | - | 62.7 | 28.7 | 4.8 | 32.8 | 66.6 | 30.9 | 22.9 |
| **MG**[2021] [61] | VGG | 4.8 | - | 59.4 | 33.6 | 5.9 | 37.5 | 69.2 | 36.8 | 26.8 |
| **SLT-Net-LT**[2022] [4] | PVTv2-B5 | 82.3 | - | 69.6 | 48.1 | 3.0 | 52.4 | 84.5 | 49.3 | 40.2 |
| **ZoomNeXt**[2024] [3] | PVTv2-B5 | 84.8 | - | 75.7 | 59.3 | 2.0 | 63.1 | 86.5 | 59.9 | 51.0 |
| **SAM2**[2024] [15] | Hiera-T | 38.9 | 1-click | 75.7 | 58.3 | 3.3 | 62.2 | 81.4 | 59.2 | 48.9 |
| **CamSAM2** | Hiera-T | 39.4 | 1-click | 77.1 | 62.2 | 3.2 | 68.1 | 83.9 | 62.6 | 50.7 |
| **SAM2**[2024] [15] | Hiera-T | 38.9 | box | 85.4 | 77.3 | 1.7 | 79.5 | 95.1 | 77.8 | 66.7 |
| **CamSAM2** | Hiera-T | 39.4 | box | **87.2** | **79.5** | **1.3** | **81.4** | **96.3** | **79.6** | **69.2** |

clearly outperforms SAM-PM, SAM2, and the fine-tuned SAM2. Even with the 1-click prompt, CamSAM2 still outperforms SAM-PM, which uses the mask prompt. The promptable methods clearly outperform other non-promptable models. Tab. 2 further compares SAM2 and CamSAM2 with the Hiera-T and Hiera-S backbones. CamSAM2 consistently outperforms SAM2 across all prompt types. With the 1-click prompt, CamSAM2 achieves mDice/mIoU gains of 12.2/9.8 with Hiera-T backbone and 13.1/12.1 with Hiera-S backbone, and further improves mIoU by 2.5/2.3 (box) and 2.6/1.0 (mask) over SAM2 on Hiera-T and Hiera-S backbones, respectively.

Fig. 3 presents qualitative comparisons on two MoCA-Mask video clips using 1-click prompts with Hiera-T as the backbone. Fig. 4 shows attention maps from the last token-to-image cross-attention layer in the mask decoder, where the SAM2 output token or the decamouflaged token acts as the query and the image embedding as key and value. Compared to SAM2, CamSAM2 produces more focused and expansive attention around target objects, highlighting its improved spatial awareness and the effectiveness of our design for VCOS. Despite CamSAM2 introducing only a marginal parameter increase of 0.5M, it delivers substantial improvements while keeping all SAM2 parameters frozen, fully preserving SAM2's original ability to segment and track objects in natural scenes.

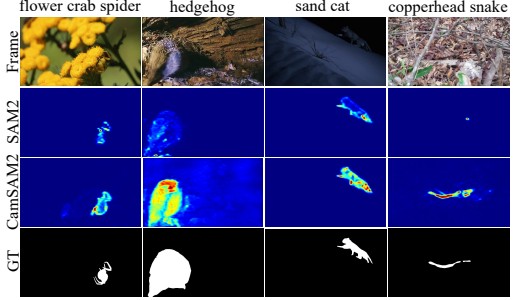

Figure 4: **Attention map visualization from SAM2 and CamSAM2 using point prompts with the Hiera-T backbone.** From *top* to *bottom*: input frames, attention with SAM2 mask token, attention with decamouflaged token, and ground-truth masks. The higher attention regions are indicated by warmer colors.

**Results on CAD.** We evaluate the zero-shot performance of CamSAM2 and SAM2 on the CAD using Hiera-T backbone with point and box prompts, as shown in Tab. 3. CamSAM2 consistently outperforms SAM2, especially with the 1-click prompt,

Table 4: **Comparisons between CamSAM2 and SAM2 on SUN-SEG-Easy and SUN-SEG-Hard.**

| Model | $S_m \uparrow$ | $F_\beta^\omega \uparrow$ | $E_m \uparrow$ | mDice $\uparrow$ |
|---|---|---|---|---|
| SUN-SEG-Easy | | | | |
| SAM2 [15] | 83.4 | 71.6 | 83.0 | 73.6 |
| CamSAM2 | **88.3** | **82.6** | **93.4** | **84.3** |
| SUN-SEG-Hard | | | | |
| SAM2 [15] | 75.5 | 58.4 | 73.4 | 61.0 |
| CamSAM2 | **86.4** | **78.2** | **91.2** | **80.6** |

Table 5: **Ablation study on the effectiveness of main components of CamSAM2.** It shows the effectiveness of each key component of CamSAM2.

| Decam. Token | IOF | EOF | OPG | mDice $\uparrow$ | mIoU $\uparrow$ |
|---|---|---|---|---|---|
| | | | | 52.1 | 44.8 |
| ✓ | | | | 54.9 | 47.0 |
| ✓ | ✓ | | | 55.2 | 47.5 |
| ✓ | ✓ | ✓ | | 55.9 | 47.9 |
| ✓ | ✓ | ✓ | ✓ | **64.3** | **54.6** |

Table 6: **Impact of using different distance metrics for $k$-means in Object Prototype Generation.** Cosine distance shows superiority.

| Distance Metric | mDice $\uparrow$ | mIoU $\uparrow$ |
|---|---|---|
| Euclidean | 61.9 | 52.7 |
| Cosine | **64.3** | **54.6** |

Table 7: **Impact of using different number of prototypes in Object Prototype Generation.**

| Prototypes ($k$) | mDice $\uparrow$ | mIoU $\uparrow$ |
|---|---|---|
| 3 | 60.2 | 51.8 |
| 5 | **64.3** | **54.6** |
| 7 | 60.6 | 50.8 |

achieving improvements of 3.4 mDice and 1.8 mIoU. With the box prompt, it also shows clear gains of 1.8 mDice and 2.5 mIoU. These results highlight CamSAM2's superior zero-shot performance, demonstrating its effectiveness for segmentation tasks with minimal user input.

**Results on SUN-SEG.** As shown in Tab. 4, CamSAM2 consistently outperforms SAM2 across all metrics on both SUN-SEG-Easy and SUN-SEG-Hard. Notably, it improves mDice by 10.7 on SUN-SEG-Easy (from 73.6 to 84.3) and by 19.6 on SUN-SEG-Hard (from 61.0 to 80.6), demonstrating strong capability in segmenting camouflaged polyps. These results confirm that CamSAM2 significantly enhances SAM2 across varying tasks of camouflage, highlighting its effectiveness and generalizability in VCOS.

### 4.3 Ablation studies

To understand the impact of each component in CamSAM2, we conduct ablation studies on MoCA-Mask using the Hiera-T backbone with the 1-click prompt. The goal is to measure the contributions of key components, including the decamouflaged token, IOF, EOF, and OPG. Additionally, we evaluated the effects of different distance metrics and prototype numbers in the OPG process.

**Impact of key components.** As shown in Tab. 5, each main component in CamSAM2 clearly contributes to its high performance. Starting from baseline (SAM2), adding the decamouflaged token alone improves mDice from 52.1 to 54.9 and mIoU from 44.8 to 47.0. Adding IOF further raises mDice to 55.2 and mIoU to 47.5. Using EOF brings mDice to 55.9 and mIoU to 47.9. With all components included, the model performs the best, achieving 64.3 on mDice and 54.6 on mIoU.

**Effect of distance metric.** We compare different distance metrics for $k$-means clustering in OPG, as shown in Tab. 6. Cosine distance performs better than Euclidean distance, likely due to its effectiveness in grouping camouflaged features by angular relationships rather than direct distances.

**Influence of number of prototypes $k$.** We examine the impact of the number of prototypes $k$, as shown in Tab. 7. The results show that both fewer or higher numbers of prototypes will reduce the performance due to *under-representation* or *redundancy*, respectively. It is observed that $k = 5$ is found to be optimal for capturing essential informative details in camouflaged features.

## 5 Conclusion

In this paper, we introduce the CamSAM2, by equipping SAM2 with the ability to accurately segment and track the camouflaged objects for VCOS. While SAM2 demonstrates strong performance across general segmentation tasks, its performance on VCOS is suboptimal due to a lack of feature optimization and architectural support for considering the challenges of VCOS. To overcome the limitations, we propose to add a learnable decamouflaged token to optimize SAM2's features for VCOS, as well as three key modules: IOF for enhancing memory-conditioned features with implicitly object-aware high-resolution features, EOF for refining features with explicit object details, and OPG for abstracting high-quality features within the object region into informative object prototypes. Our experiments on three popular benchmarks of two camouflaged scenarios demonstrate that CamSAM2 clearly improves VCOS performance over SAM2, especially with point prompts, while fully inheriting SAM2's zero-shot capability. By setting new state-of-the-art performance, CamSAM2 offers a more practical and effective solution for real-world VCOS applications.

## Acknowledgments

This project was partially supported by grant #2022-279 of the Strategic Focus Area "Personalized Health and Related Technologies (PHRT)" of the ETH Domain (Swiss Federal Institutes of Technology).

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

# A    Appendix

## A.1    Training details

During the training process, each training video clip consists of 8 frames, the input frames are resized to $1024 \times 1024$, and the ground truths are resized to $256 \times 256$ since the raw predicted logits are 1/4 of the original size. We train CamSAM2 with a batch size of 4. The predicted final mask is interpolated to $1024 \times 1024$ and then encoded with the visual feature of the current frame as the memory feature in the memory bank for subsequent frames.

## A.2    Comparative analysis across backbones and prompt types

Table 8: **Detailed comparisons between CamSAM2 and SAM2 on MoCA-Mask across various backbones and prompt types.** CamSAM2 *consistently* outperforms SAM2 for all cases. Improvements are highlighted in dark green.

| Model | Prompt | mDice ↑ | mIoU ↑ |
|---|---|---|---|
| **Hiera-T** | | | |
| SAM2
CamSAM2 | 1-click | 52.1
64.3 (+12.2) | 44.8
54.6 (+9.8) |
| SAM2
CamSAM2 | box | 72.7
75.5 (+2.8) | 62.3
64.8 (+2.5) |
| SAM2
CamSAM2 | mask | 77.1
80.2 (+3.1) | 67.9
70.5 (+2.6) |
| **Hiera-S** | | | |
| SAM2
CamSAM2 | 1-click | 54.9
68.0 (+13.1) | 46.7
58.8 (+12.1) |
| SAM2
CamSAM2 | box | 73.7
76.4 (+2.7) | 63.8
66.1 (+2.3) |
| SAM2
CamSAM2 | mask | 80.3
81.4 (+1.1) | 70.7
71.7 (+1.0) |
| **Hiera-B+** | | | |
| SAM2
CamSAM2 | 1-click | 55.5
68.0 (+12.5) | 45.5
58.4 (+12.9) |
| SAM2
CamSAM2 | box | 73.0
75.1 (+2.1) | 63.1
64.9 (+1.8) |
| SAM2
CamSAM2 | mask | 77.8
81.0 (+3.2) | 68.5
71.2 (+2.7) |
| **Hiera-L** | | | |
| SAM2
CamSAM2 | 1-click | 69.3
73.5 (+4.2) | 55.8
63.3 (+7.5) |
| SAM2
CamSAM2 | box | 74.9
76.2 (+1.3) | 65.4
66.5 (+1.1) |
| SAM2
CamSAM2 | mask | 80.9
81.9 (+1.0) | 71.1
72.1 (+1.0) |

Tab. 8 compares the performance of SAM2 and CamSAM2 on the MoCA-Mask dataset across various backbones (Hiera-T, Hiera-S, Hiera-B+, and Hiera-L) and prompt types (click, box, and mask). The results consistently demonstrate that CamSAM2 outperforms SAM2 across all scenarios, with improvement varying depending on the prompt type and backbone.

For click prompts, CamSAM2 achieves substantial improvements, particularly with smaller backbones such as Hiera-T and Hiera-S. For instance, with Hiera-T, CamSAM2 achieves a significant improvement of 12.2 mDice and 9.8 mIoU, while with Hiera-S, the gains are even larger at 13.1

mDice and 12.1 mIoU. These results highlight CamSAM2's ability to utilize minimal user input to enhance segmentation performance effectively.

For box prompts, CamSAM2 consistently maintains its advantage. With the Hiera-T backbone, CamSAM2 achieves gains of 2.8 mDice and 2.5 mIoU. With Hiera-L backbone, it delivers consistent improvements.

For mask prompts, where the input is the most detailed, CamSAM2 continues to outperform SAM2. For example, with Hiera-T, CamSAM2 achieves gains of 3.1 mDice and 2.6 mIoU, while with Hiera-B+, the improvements are 3.2 mDice and 2.7 mIoU.

Overall, the results highlight CamSAM2's adaptability across different backbones and prompt types. Its ability to achieve notable improvements with minimal input, while consistently maintaining advantages as prompts become more detailed, underscores its practicality and effectiveness for real-world applications requiring accurate segmentation of camouflaged objects in videos.

### A.3 Comparative analysis across point prompt with different numbers of clicks

We evaluate the point prompt by randomly selecting points from the ground-truth mask regions using a fixed random seed of $42$. Click prompts are evaluated on the MoCA-Mask dataset with varying numbers of clicks: 1, 2, 3, and 5. Across all settings, CamSAM2 consistently outperforms SAM2, as shown in Tab. 9, demonstrating its robustness and adaptability.

With the 2-click prompt, CamSAM2 achieves significant improvements of 16.4 mDice and 15.2 mIoU with the Hiera-T backbone. Similarly, with the Hiera-S backbone, it achieves gains of 6.8 mDice and 5.9 mIoU, effectively leveraging additional user input for enhanced segmentation accuracy. For the 3-click prompt, CamSAM2 continues to deliver improvements, achieving a 3.4 mDice gain with the Hiera-T backbone and a 1.9 mDice gain with the Hiera-S backbone. These results showcase CamSAM2's ability to utilize increasing user input effectively, maintaining its advantage in producing accurate segmentation outcomes.

With the 5-click prompt, while both models benefit from the additional user input, CamSAM2 still achieves noticeable gains. The Hiera-T backbone records an improvement of 2.2 mDice and 1.9 mIoU, while with the Hiera-S backbone, the gains are 5.4 mDice and 4.6 mIoU, highlighting CamSAM2's consistent effectiveness across varying input levels. These results emphasize CamSAM2's ability to adapt and maintain robust performance as the complexity of the interaction grows.

These results highlight CamSAM2's adaptability and consistent effectiveness across the varying number of user clicks. Its ability to achieve substantial improvements, while maintaining robust performance, underscores its practicality and scalability for real-world applications that demand precise segmentation of VCOS.

### A.4 Comparison with SAM2-Adapter

To provide a fair and comprehensive comparison, we evaluate CamSAM2 alongside SAM2-Adapter [38]. SAM2-Adapter is an approach to adapt SAM2 to downstream tasks and achieve enhanced performance. This method effectively integrates task-specific knowledge with the general knowledge learned by the model. During this process, only the adaptor layers and the mask decoder are trained, while the rest of the image encoder is kept frozen. Since the official SAM2-Adapter implementation uses Hiera-L as the backbone, we directly compare on the same backbone to avoid unintended modifications to their codebase. We re-train the SAM2-Adapter on the MoCA-Mask dataset using their provided settings. Results are reported in Tab. 10. Importantly, CamSAM2 and SAM2-Adapter represent two fundamentally different design philosophies for extending SAM2:

**Evaluation granularity.** SAM2-Adapter operates at the *image level*, making per-frame predictions independently. In contrast, CamSAM2 is designed for *video level* inference, where only the first frame receives a prompt, and subsequent frames are segmented using memory-augmented features that evolve over time. This enables prototype refinement and accumulated improvements across frames, which is a capability SAM2-Adapter lacks.

**Prompt usage.** SAM2-Adapter relies on a *learned prompt* injected into different stages in the image encoder at every image during inference. In contrast, CamSAM2 follows the original SAM2 design

Table 9: **Detailed comparisons between CamSAM2 and SAM2 with Hiera-T and Hiera-S backbones on MoCA-Mask using different numbers of click prompts.**

| Model | Prompt | mDice ↑ | mIoU ↑ |
|---|---|---|---|
| Hiera-T | | | |
| SAM2
CamSAM2 | 1-click | 52.1
64.3 (+12.2) | 44.8
54.6 (+9.8) |
| SAM2
CamSAM2 | 2-click | 48.4
64.8 (+16.4) | 39.8
55.0 (+15.2) |
| SAM2
CamSAM2 | 3-click | 63.7
67.1 (+3.4) | 55.4
56.3 (+0.9) |
| SAM2
CamSAM2 | 5-click | 65.9
68.1 (+2.2) | 56.7
58.6 (+1.9) |
| Hiera-S | | | |
| SAM2
CamSAM2 | 1-click | 54.9
68.0 (+13.1) | 46.7
58.8 (+12.1) |
| SAM2
CamSAM2 | 2-click | 56.8
63.6 (+6.8) | 47.8
53.7 (+5.9) |
| SAM2
CamSAM2 | 3-click | 71.7
73.6 (+1.9) | 61.7
62.9 (+1.2) |
| SAM2
CamSAM2 | 5-click | 68.5
73.9 (+5.4) | 58.4
63.0 (+4.6) |

Table 10: **Comparison between SAM2-Adapter and CamSAM2 with Hiera-L as the backbone.**

| Model | Prompt | $S_m$ ↑ | $F_\beta^\omega$ ↑ | MAE ↓ | $F_\beta$ ↑ | $E_m$ ↑ | mDice ↑ | mIoU ↑ |
|---|---|---|---|---|---|---|---|---|
| SAM2-Adapter [38] | - | 68.4 | 38.7 | 0.9 | 43.0 | 80.0 | 43.2 | 35.8 |
| CamSAM2 | 1-click | **82.2** | **71.8** | **0.6** | **73.4** | **92.1** | **73.5** | **63.3** |

as a *promptable segmentation model*, accepting a single external prompt (e.g., a click) on the first frame and leveraging internal mechanisms to propagate the segmentation over time. While our usage of ground-truth-derived prompts may raise fairness concerns, this is a common and practical setting in semi-supervised video segmentation and is consistent with SAM2's prompt-driven paradigm.

In summary, CamSAM2 and SAM2-Adapter offer two distinct solutions. CamSAM2 enhances SAM2 for video by leveraging memory and minimal user guidance, while SAM2-Adapter adapts SAM2 to images through internal prompt learning. Although they address different use cases, our results demonstrate that CamSAM2 provides a more effective and temporally coherent framework for VCOS, maintaining compatibility with SAM2's promptable design without modifying its parameters.

### A.5 Additional training and inferencing details

As we mentioned in the §4.1, we adopt a combined loss of binary cross-entropy (BCE) and dice loss for mask predictions across the entire video sequence. This loss applies to both the original SAM2 mask logits $\mathbf{R}_i$ and our CamSAM2 mask logits $\mathbf{R}_i^c$. Although the original SAM2 parameters remain frozen during training, we still apply supervision to its outputs. The reason is that the introduction of our learnable decamouflaged token changes the self-attention dynamics in the mask decoder: this token is concatenated with the original output tokens and jointly participates in self-attention and cross-attention layers (see Fig. 2(a) in the main paper). Consequently, the outputs $\mathbf{R}_i$ are no longer guaranteed to be identical to those of the original SAM2. By supervising them, we ensure that SAM2's predictions remain aligned with the segmentation intent and are not perturbed by the token interactions.

Inspired by HQ-SAM [62], we adopt a simple post-processing strategy where the final mask prediction is obtained by averaging SAM2's mask logits $R_t$ and the CamSAM2 mask logits $R_t^c$, which is calculated by the SAM2 mask token and the decamouflaged token. This strategy integrates SAM2's global representation with the fine-grained, camouflage-aware features extracted by our proposed

modules. Despite its simplicity, this approach effectively unifies global generalization and fine-grained object sensitivity.

Tab. 11 reports mIoU scores on the Hiera-T backbone. The first column shows the original SAM2 performance. The second and third columns report the mask predictions from the SAM2 token and the decamouflaged token within CamSAM2, respectively. The final column presents the result of averaging both. Each individual output mask already outperforms the SAM2 baseline, and their combination leads to further improvements across all prompt types, validating the effectiveness of our averaging strategy.

Table 11: **Comparison of SAM2, individual token predictions within CamSAM2, and their combination.** Results are reported as mIoU on the Hiera-T backbone across three prompt types. "SAM2 baseline" refers to the vanilla SAM2 prediction. "SAM2 token" denotes the prediction generated from the SAM2 mask token, while "decamouflaged token" refers to the prediction from the decamouflaged token after integrating features from our proposed IOF, EOF, and OPG modules. "CamSAM2" is the final result obtained by averaging the mask logits calculated by the SAM2 token and the decamouflaged token.

| Prompt | SAM2 baseline | SAM2 token | decamouflaged token | CamSAM2 |
|---|---|---|---|---|
| 1-click | 44.8 | 53.3 | 53.1 | **54.6** |
| box | 62.3 | 63.9 | 64.1 | **64.8** |
| mask | 67.9 | 70.3 | 68.4 | **70.5** |

Notably, the mask of the SAM2 output token within CamSAM2 yields better results than in the original SAM2 model, despite having identical parameters and no additional fine-tuning. This improvement stems from two key factors. First, all output tokens, including the SAM2 mask tokens and the decamouflaged token, are jointly processed in the same transformer, where they interact through a self-attention layer and different cross-attention layers. Second, SAM2's architecture conditions the current frame's feature $F_t^{mem}$ on the previously predicted mask stored in memory, enriching the encoder's output over time. This mutual influence further enhances their individual representational capacity.

## A.6 First frame performance analysis

We evaluate CamSAM2 on the first frame of each video clip on the MoCA-Mask dataset using a 1-click point prompt with Hiera-T as the backbone. This setting isolates the model's ability to handle camouflaged object segmentation, as on the first frame, the OPG module is naturally inactive, since there are no prior frames from which to extract prototypes. This makes the first frame an ideal testbed for evaluating the model's segmentation capability only without temporal information.

Table 12: **First frame comparison between SAM2 and CamSAM2 with the point prompt.**

| Model | $S_m \uparrow$ | $F_\beta^\omega \uparrow$ | MAE $\downarrow$ | $F_\beta \uparrow$ | $E_m \uparrow$ | mDice $\uparrow$ | mIoU $\uparrow$ |
|---|---|---|---|---|---|---|---|
| SAM2 | 72.2 | 56.7 | 7.4 | 58.1 | 75.9 | 58.6 | 51.1 |
| CamSAM2 | **73.0** | **59.4** | **7.2** | **60.8** | **80.7** | **61.4** | **52.6** |

As shown in Tab. 12, CamSAM2 consistently outperforms SAM2 across all metrics on the first frame. This improvement is particularly meaningful, as it demonstrates that CamSAM2 enhances SAM2's *segmentation* capability even in the absence of temporal cues, laying a stronger foundation for subsequent predictions. Since CamSAM2 stores the predicted mask of each frame into memory and uses it to generate object prototypes, these prototypes can progressively enhance segmentation in future frames within EOF, resulting in cumulative performance gains over time, which shows that CamSAM2 further enhances the *tracking* ability of SAM2 for the VCOS task.

## A.7 Efficiency analysis of OPG

To assess the runtime efficiency of CamSAM2, we profile its per-frame latency using the Hiera-T backbone on a representative 106-frame camouflaged video from the MoCA-Mask test set. Tab. 13

Table 13: **Average runtime per frame and overhead breakdown of CamSAM2.**

| Component | Avg Time (ms) | Relative Percentage |
|---|---|---|
| SAM2 | 83.7 | 100.0% |
| FPS | 3.5 | 4.2% |
| k-means | 1.0 | 1.2% |
| OPG | 4.6 | 5.5% |
| CamSAM2 | 89.7 | 107.2% |

reports the average runtime breakdown and compares CamSAM2 against the baseline SAM2. Overall, CamSAM2 introduces only 6 ms (7.2%) additional latency per frame relative to SAM2, demonstrating that the method remains efficient. The OPG module, which consists of FPS sampling and 1-iter $k$-means clustering, accounts for the majority of the runtime overhead. All other added components, including IOF, EOF, and associated processing, together contribute approximately 1.4 ms (1.7%) per frame.

We profile the computational complexity in Tab. 14. CamSAM2 introduces an additional 14.4 GFLOPs, which is independent of different backbone architectures. Based on the Hiera-T backbone, the total FLOPs increase from 137.2 GFLOPs to 151.5 GFLOPs, which corresponds to approximately a 10% increase. This added cost results in a significant performance gain of 9.8 mIoU with point prompts. Moreover, when using the Hiera-L backbone, which image encoder requires 810 GFLOPs, the same 14.4 GFLOPs overhead accounts for only a 1.7% increase in total FLOPs. Even with such a small relative increase, CamSAM2 still achieves a notable 7.5 mIoU gain.

Table 14: **FLOPs comparison between SAM2 and CamSAM2.**

| Module | FLOPs (G) |
|---|---|
| Image encoder | 103.0 |
| Memory encoder | 5.0 |
| Memory attention | 27.4 |
| Mask decoder | 1.8 |
| **SAM2** | **137.2** |
| CamSAM2 extra | 14.4 |
| **CamSAM2** | **151.5** |

These findings confirm that CamSAM2 remains efficient and is suitable for practical deployment, even in time-sensitive VCOS applications.

### A.8 Additional analysis and ablation studies

**Number of Decamouflaged Tokens.** We conduct an ablation study on the number of decamouflaged tokens to evaluate its effect on representation quality. As shown in Tab. 15, increasing the number of tokens from 1 to 3 results in a comparable performance in mIoU. This indicates that adding more tokens does not structurally improve the representation quality, but may introduce redundancy and additional computational cost.

Table 15: **Impact of using different numbers of decamouflaged tokens.**

| Model | Number of Decam. Token | mIoU |
|---|---|---|
| SAM2 | - | 44.8 |
| CamSAM2 | 3 | 53.3 |
| CamSAM2 | 1 | **54.6** |

**Sampling and Clustering Strategies.** To evaluate the robustness and generality of the OPG module, we experiment with different strategies for sampling and clustering. Specifically, we compare (i)

Farthest Point Sampling (FPS) versus Average Sampling for selecting initial cluster centers, and (ii) $k$-means versus Gaussian Mixture Model (GMM) for clustering. Results are reported in Tab. 16. All combinations yield reasonable performance, confirming the flexibility of OPG. Notably, FPS better preserves spatial diversity among selected features, leading to more representative prototypes.

Table 16: **Impact of different sampling and clustering strategies in OPG.**

| Sampling | Clustering | mIoU |
|---|---|---|
| Average Sampling | $k$-means | 52.1 |
| Average Sampling | GMM | 52.7 |
| FPS | GMM | 53.6 |
| FPS | $k$-means | **54.6** |

These findings show that OPG is not overly sensitive to specific implementation choices, while FPS sampling in combination with $k$-means clustering offers the best trade-off between representativeness and segmentation accuracy.

## A.9 Architecture toggle between CamSAM2 and SAM2

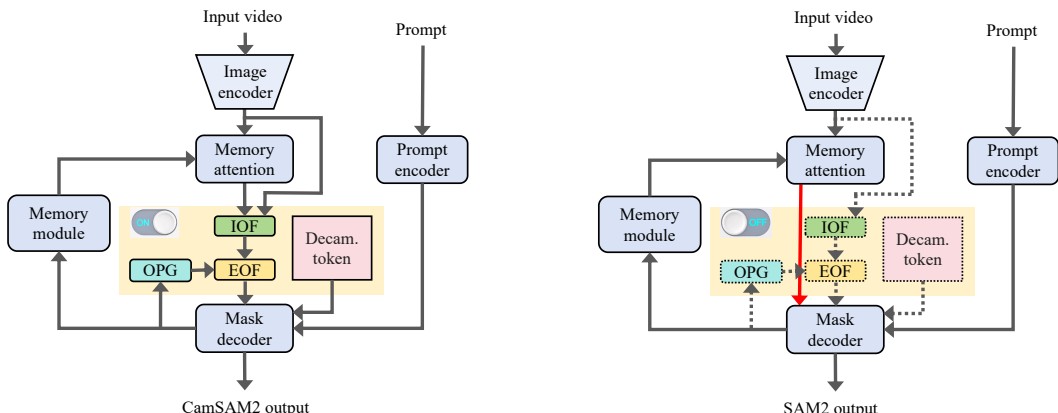

Figure 5: **Illustration of the architecture toggle**. The toggle switch enables or disables the proposed modules for VCOS containing the decamouflaged token, IOF, EOF, and OPG. Modules and flows in dashed lines indicate the disabled state.

To better illustrate the relationship between CamSAM2 and the original SAM2 pipeline, Fig. 5 presents a side-by-side comparison. When the toggle is set to **OFF**, the proposed modules are completely bypassed, and the pipeline reverts to the SAM2 architecture. Importantly, since all SAM2 parameters are fixed during the training of CamSAM2, the model behaves identically to SAM2 in this mode, preserving its original structure and performance across various tasks. This design ensures compatibility and flexibility, allowing seamless integration without disrupting SAM2's performance.

## A.10 Limitation

CamSAM2 is built upon the SAM2 architecture, inheriting both its strengths and limitations. While our proposed modifications significantly improve SAM2's performance in VCOS tasks, we do not alter the core architecture of SAM2. As a result, CamSAM2 retains several of SAM2's known limitations. Specifically, it does not address SAM2's challenges in handling shot changes and long occlusions. Future work may consider integrating explicit motion information or relational modeling to further improve the robustness of the model.

## A.11 Interpreting prototypes via cosine similarity maps

To improve interpretability, we visualize the cosine similarity between the object prototype from preceding frames and the current-frame feature map of two selected videos. As shown in Fig. 6, the

similarity maps consistently highlight semantically meaningful object regions while suppressing the background, indicating that the prototypes abstract object-aware semantics and remain temporally consistent rather than drifting to spurious parts.

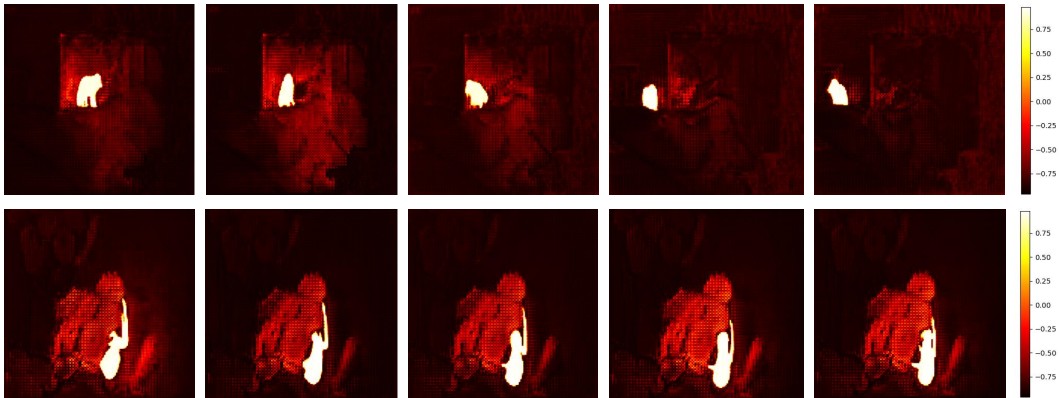

Figure 6: **The cosine similarity map between the preceding frames prototype and the current frame feature map.**

## A.12 More qualitative result visualization

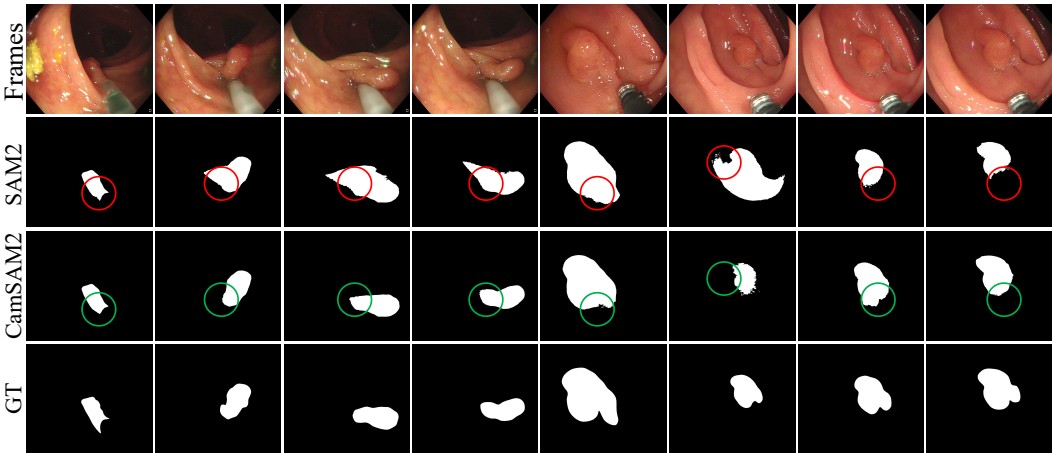

Figure 7: **Qualitative comparisons between SAM2 and CamSAM2 using mask prompt with the Hiera-T backbone on two video clips of SUN-SEG-Hard.** From *top* to *bottom*: the input frames, SAM2's results, CamSAM2's results, and ground-truth masks. CamSAM2 demonstrates improved accuracy in segmenting camouflaged polyps, as shown by the circles. *Best viewed in color.*

Fig. 7 presents two qualitative example video clips of SUN-SEG-Hard, comparing the segmentation performance of SAM2 and CamSAM2 using mask prompts with the Hiera-T backbone. Compared with SAM2, CamSAM2 can segment polyps more accurately, including more precise regions and boundaries.

Fig. 8 and Fig. 9 present more comprehensive qualitative results for six video clips in MoCA-Mask, comparing the segmentation performance of SAM2 and CamSAM2 using 1-click point prompts with the Hiera-T backbone. The results clearly demonstrate CamSAM2's advantages in segmenting camouflaged objects in videos. For instance, in the first (*stick insect*) and second (*rusty spotted cat*) examples from Fig. 8, SAM2 is able to segment parts of the camouflaged objects; however, CamSAM2 significantly outperforms by successfully segmenting larger and more complete regions of the objects. Furthermore, in the third (*pygmy seahorse*) example, SAM2 produces an incorrect segmentation, failing to segment the camouflaged object accurately, while CamSAM2 still manages to segment the

correct regions of the object. These findings highlight CamSAM2's ability to both enhance accuracy in cases where SAM2 performs moderately well and to maintain reliable segmentation performance in challenging scenarios where SAM2 fails. This underscores CamSAM2's robustness and effectiveness in tackling diverse and complex tasks in VCOS.

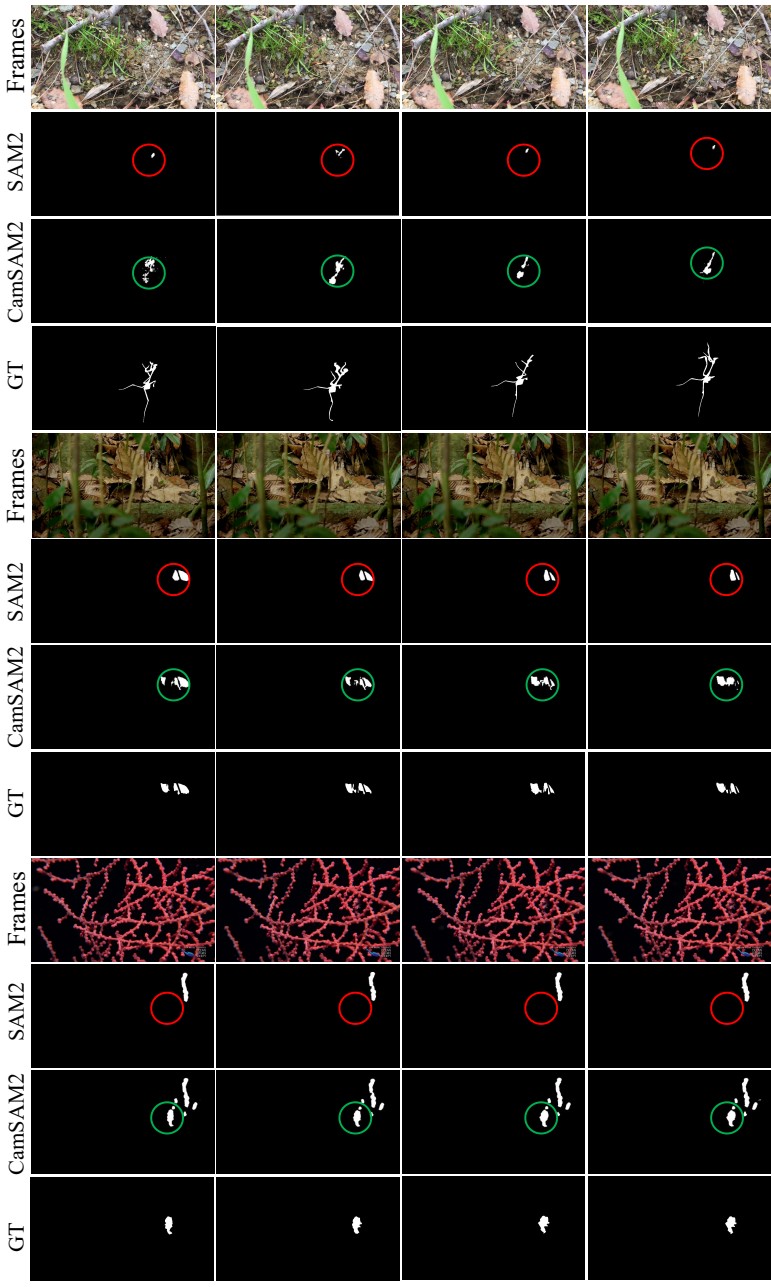

Figure 8: **More qualitative comparisons between SAM2 and CamSAM2 using 1-click point prompt with the Hiera-T backbone on three video clips.** From *top* to *bottom*: the input frames, SAM2's results, CamSAM2's results, and ground-truth masks. CamSAM2 demonstrates improved accuracy in segmenting camouflaged objects, as shown by the circles. *Best viewed in color.*

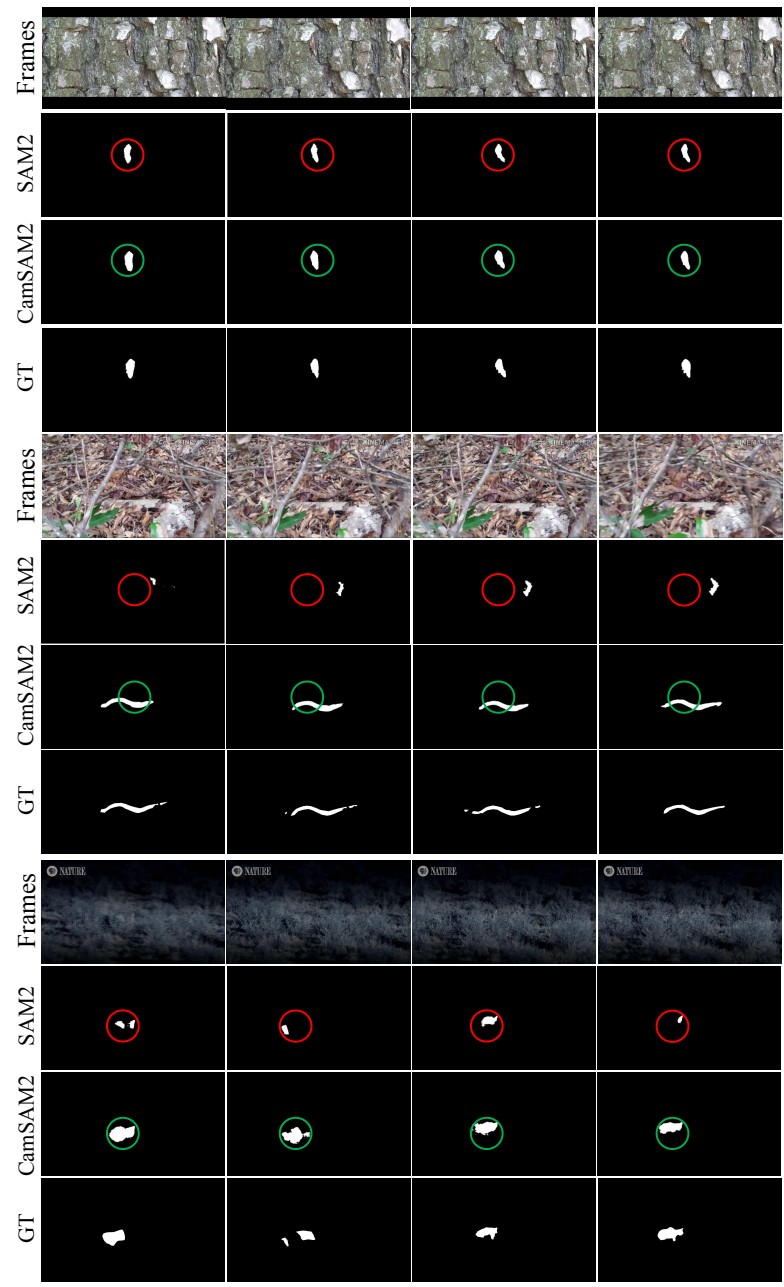

Figure 9: **More qualitative comparisons between SAM2 and CamSAM2 using 1-click point prompt with the Hiera-T backbone on three video clips.** *Best viewed in color.*

