# OpenReview forum: "CamSAM2: Segment Anything Accurately in Camouflaged Videos"
_NeurIPS.cc/2025/Conference — NeurIPS 2025 poster_

### Official Review · Reviewer_L184 · 2025-06-22

**Clarity:** 3
**Significance:** 3
**Originality:** 3
**Rating:** 5
**Confidence:** 5

**Summary:**

This paper proposes a segmentation method based on SAM2 for camouflaged object segmentation in videos, named CamSAM2. Specifically, the authors introduce a decamouflaged token to provide the flexibility of feature adjustment for VCOS. To make full use of fine-grained and high-resolution features from the current frame and previous frames, the authors propose implicit object-aware fusion (IOF) and explicit object-aware fusion (EOF) modules, respectively. Extensive experimental results validate the effectiveness of the CamSAM2.

**Questions:**

1. Although the paper has already been compared with several existing methods, the baseline methods for the comparison mainly focus on specific covert video segmentation tasks. For some general video segmentation methods (such as Mask R-CNN, DeepLab, etc.), although they may perform poorly in the covert video segmentation task, comparing these methods can more comprehensively demonstrate the advantages of CamSAM2.
2. From the ablation experiments, it can be seen that compared with the IOF and EOF modules, the addition of the Decam. Token and OPG modules in the model result in a significant performance improvement. However, the key roles of these components are not mentioned in the paper.

**Ethical Concerns:**

["NO or VERY MINOR ethics concerns only"]

**Final Justification:**

After carefully evaluating the author's rebuttal, I recommend Accept.

**Limitations:**

The authors should provide the computational complexity of various algorithms and inference speed to more comprehensively evaluate the method proposed in this paper.

**Quality:**

3

**Strengths And Weaknesses:**

1. Strengths:
(1) The research motivation is clear, and the proposed method is easy to follow.
(2) Based on the SAM2 framework, the authors designed the IOF and EOF modules to segment camouflaged objects in videos.

2. Weaknesses:
(1) The description of some key modules (such as IOF, EOF, and OPG) in the paper is relatively concise, and some details may not be clear enough. In the OPG module, although Farthest Point Sampling and k-means are mentioned, there is no detailed explanation on how to select the value of k and how to handle the prototype consistency between different frames.
(2) Although CamSAM2 has achieved significant performance improvements, the paper does not mention the real-time capability of the model. For some practical applications (such as wildlife monitoring, search and rescue, etc.), real-time performance is an important consideration factor.
(3) The third chapter is dedicated to the methods section, but 3.6 (training and inference strategies) and 3.7 (experimental setup) are related to the experiments.
(4) For Table 1, it is suggested that the years of the comparative test models be listed.

---

> ### Author Rebuttal · Authors · 2025-07-31
>
> We sincerely thank you for the positive feedback and thoughtful suggestions. We are glad that you found our research motivation to be clear and our method design to be easy to follow. We also appreciate your recognition of the architectural clarity and the integration of the IOF and EOF modules based on the SAM2 framework. Your comments encourage us to further refine and clarify the technical aspects of our work, and we address your insightful questions and concerns below.
>
> ---
>
> ## W1: OPG details clarification
> Thanks for your thoughtful questions.
> 1. Regarding the number of prototypes ($k$), we have investigated its impact in Table 7 in the main paper. The results show that both fewer or higher numbers of prototypes will reduce the performance due to under-representation or redundancy, but are still effective in abstracting prototypes.
> 2. Regarding the consistency of prototypes across time, these prototypes remain consistent in a semantic sense, as they are all abstracted from the previously predicted camouflaged regions. Since these regions represent the same object over time, the resulting prototypes serve as temporally coherent and semantically aligned for guiding the segmentation in subsequent frames.
>
> To improve interpretability, we calculate cosine similarity maps between each prototype of previous frames and the feature map of the current frame. The visualizations reveal that different prototypes consistently activate on semantically meaningful object regions. For example, the prototype shows strong activations (high similarity) precisely on the camouflaged object, while suppressing the background. This suggests that the prototypes successfully abstract object-aware semantics and are consistent across time. We will present detailed visualizations and analysis in the final version of the paper to further support interpretability.
>
> ---
>
> ## W2: Real-time capability concern
> Thanks for raising this practical concern. We profile the runtime overhead of CamSAM2 on a 106-frame video and report the average per-frame latency in Table 1. CamSAM2 adds only 6 ms latency per frame on top of SAM2, resulting in a total per-frame time of 89.7 ms. This demonstrates the model's practicality in real-time applications such as wildlife monitoring, search and rescue.
>
>
> | Component   | Avg Time (ms) | Relative Percentage |
> |:-------------|:---------------:|:---------------------:|
> | SAM2        | 83.7          | 100.0%              |
> | FPS     | 3.5           | 4.2%                |
> | k-means | 1.0           | 1.2%                |
> | OPG         | 4.6           | 5.5%                |
> | CamSAM2     | 89.7          | 107.2%              |
>
> **Table 1. Average runtime per frame and overhead breakdown of CamSAM2 compared to original SAM2.**
>
> ---
>
> ## W3: Structural issue with Section 3
> Thanks for your suggestion. We agree that Sections 3.6 and 3.7 are more appropriately arranged under the Experiments section. We will move them into Section 4 in the final version to improve structural consistency.
>
> ---
>
> ## W4: Publication year in Table
> Thanks for your advice. We will add the publication year in the final version.
>
> ---
>
> ## Q1: More VOS models comparisons
> Thanks for your suggestion. We compare CamSAM2 with two recent and representative SOTA video object segmentation (VOS) models: Cutie [1] and XMem++ [2], both widely used in VOS tasks. For fairness, we evaluate all models using the ground truth of the first frame as the mask prompt on the MoCA-Mask. As shown in Table 2, CamSAM2 achieves 70.5 mIoU, surpassing all baselines with comparable model sizes. This demonstrates the strong performance of CamSAM2 over both VOS-specific models and its SAM2-based baseline.
>
>
>
> | Method      | Params (M) | mIoU |
> |:-------------|:---------------:|:---------------------:|
> | Cutie-small | 28.7       | 60.8 |
> | XMem++      | 62.2       | 61.4 |
> | SAM2        | 38.9       | 67.9 |
> | CamSAM2 | 39.4       | **70.5** |
>
> **Table 2. Comparison with VOS models.**
>
> ---
>
> ## Q2: Highlighting the roles of components
> We appreciate your suggestion. In CamSAM2, the decamouflaged token and OPG module are the core components that enable our method to adapt SAM2 for the VCOS task. The decamouflaged token allows the model to learn camouflaged features for camouflaged objects and is used to generate the decamouflaged mask without modifying SAM2's architecture, while OPG abstracts object-aware prototypes from previous frames to guide segmentation across time. The IOF and EOF modules play supporting roles, further enhancing spatial and temporal feature representation. To make this distinction clearer, we will revise the final version to more explicitly highlight the relative importance and motivation behind each module.
>
> ---
>
> ## Limitation
> Thanks for raising this important point. We agree that both computational complexity and inference speed are critical considerations. As mentioned in W2, inference speed has already been addressed. Besides, we profile the computational complexity in Table 3. CamSAM2 introduces an additional 14.4 GFLOPs, which is independent of different backbone architectures. Based on the Hiera-T backbone, the total FLOPs increase from 137.2 GFLOPs to 151.5 GFLOPs, which corresponds to approximately a 10% increase. This added cost results in a significant performance gain of 9.8 mIoU with point prompts.
> Moreover, when using the Hiera-L backbone, which image encoder requires 810 GFLOPs, the same 14.4 GFLOPs overhead accounts for only a 1.7% increase in total FLOPs. Even with such a small relative increase, CamSAM2 still achieves a notable 7.5 mIoU gain. This highlights the efficiency of our design: the added modules contribute strong segmentation improvements at minimal computational cost.
>
>
>
> | Module          | FLOPs (G) |
> |:-------------|:---------------:|
> | Image Encoder   | 103.0     |
> | Memory Encoder  | 5.0       |
> | Memory Attention| 27.4      |
> | Mask Decoder    | 1.8       |
> | **SAM2**        | **137.2** |
> | CamSAM2 modules| 14.4      |
> | **CamSAM2**     | **151.5** |
>
> **Table 3. FLOPs comparison between SAM2 and CamSAM2.**
>
> ---
>
> References
> [1] Putting the object back into video object segmentation.
> [2] Xmem++: Production-level video segmentation from few annotated frames.

---

> > ### Author Response · Authors · 2025-08-04
> >
> > Dear Reviewer,
> >
> > Thank you very much for your thoughtful feedback on our submission. We have carefully addressed your comments in our response.
> >
> > If you have any additional questions or concerns, we would be more than happy to clarify.
> >
> > We sincerely appreciate your time and efforts in reviewing our work.

---

> > ### Comment · Reviewer_L184 · 2025-08-07
> > **The authors have comprehensively and convincingly addressed my concerns. Their added experiments, visualizations, and structural adjustments fully resolve the issues I raised. I recommend accepting their revisions and including the promised details in the final manuscript.**
> >
> > 1. OPG Details (W1)
> > The authors supplemented Table 7 with ablation results on the number and consistency of prototypes and provided cosine similarity visualizations between prototypes and current feature maps. These clarify that prototypes remain semantically coherent over time and accurately activate target regions, improving interpretability.
> > 2. Real-time Capability (W2)
> > They profiled CamSAM2’s runtime overhead on a 106-frame video, reporting an average per-frame latency of 89.7 ms (6 ms additional to SAM2) along with module-wise breakdowns. This demonstrates the method’s practical real-time performance for wildlife monitoring and similar applications.
> > 3. Section Structure Adjustment (W3)
> > The authors agreed to relocate Sections 3.6 and 3.7 under the Experiments section, enhancing the paper’s structural consistency.
> > 4. Adding Publication Year (W4)
> > They commit to including publication years of comparison models in the final version.
> > 5. Additional VOS Model Comparisons (Q1)
> > In Table 2, they compare CamSAM2 against Cutie-small and XMem++—two representative state-of-the-art VOS methods—as well as SAM2, showing CamSAM2 achieves 70.5 mIoU on MoCA-Mask, illustrating competitive performance.
> > 6. Highlighting Component Roles (Q2)
> > The rebuttal elaborates on the design motivations and relative importance of the Decam token, OPG, IOF, and EOF modules, making their contributions to the VCOS task clearer.
> > 7. Computational Complexity (Limitation)
> > Table 3 quantifies FLOPs for each component and compares total FLOPs of SAM2 (137.2 G) vs. CamSAM2 (151.5 G, ~10% increase), further noting that with a larger Hiera-L backbone, the overhead proportion decreases. This validates the efficiency–performance tradeoff.

---

> > > ### Author Response · Authors · 2025-08-07
> > >
> > > Dear Reviewer,
> > >
> > > We are glad to hear that our response has comprehensively and convincingly addressed your concerns and fully resolved the issues you raised. Your comments were extremely helpful in improving the clarity and structure of our paper. We will incorporate the added experiments, visualizations, and structural adjustments in the final version of the paper. We sincerely appreciate your thoughtful feedback and the time you dedicated to reviewing our work!

---

> > > > ### Comment · Reviewer_L184 · 2025-08-07
> > > > **Code is expected**
> > > >
> > > > Thank you for your reply. We have reviewed your relevant paper on arXiv. If you can provide the corresponding code on this platform, and its accuracy and effectiveness are verified through testing, we will adjust the score from 3 to 5.

---

> > > > > ### Author Response · Authors · 2025-08-08
> > > > >
> > > > > Dear Reviewer,
> > > > >
> > > > > Following your request, we have released the code and trained models in an anonymous link: https://anonymous.4open.science/r/CamSAM2-anonymous-57D3. If you have other concerns, please kindly let us know.

---

> > > > > ### Author Response · Authors · 2025-08-08
> > > > >
> > > > > Dear Reviewer,
> > > > >
> > > > > As the rebuttal period is about to end, we would like to check if you encounter any issues with running or understanding our code. If there is anything unclear or if you have specific questions, please feel free to let us know so we can address them promptly.
> > > > >
> > > > > Thank you for your time and effort in reviewing our work.

---

> > > > > ### Author Response · Authors · 2025-08-09
> > > > >
> > > > > Dear Reviewer,
> > > > >
> > > > > We sincerely appreciate your constructive feedback.
> > > > >
> > > > > As you mentioned, our rebuttal has comprehensively and convincingly addressed your concerns. We have also released the code as per your request.
> > > > > Since the rebuttal will end in a few hours, we would like to ask if you could provide a final rating for our work.

---

### Official Review · Reviewer_rr2s · 2025-06-30

**Clarity:** 3
**Significance:** 3
**Originality:** 3
**Rating:** 5
**Confidence:** 4

**Summary:**

In this paper, the authors address the suboptimal performance of SAM2 in Video Camouflaged Object Segmentation (VCOS), particularly when provided with simple prompts like points or boxes. They propose Camouflaged SAM2 (CamSAM2), a method that enhances SAM2’s capabilities for camouflaged scenes without altering its original parameters. Specifically, CamSAM2 introduces an implicit object-aware fusion (IOF) module, an explicit object-aware fusion (EOF) module, and an object prototype generation (OPG) mechanism. Extensive experiments are presented to demonstrate the effectiveness of the proposed approach.

**Questions:**

1) I question why the authors compute loss twice in their training strategy. If CAMSAM2's prediction map can directly calculate loss with the ground truth, what is the benefit of computing an additional loss with SAM2? The authors have not discussed this design choice in their ablation study.
2) The authors discuss using box and 1-click prompts in their experiments. However, we know that camouflaged object detection aims to find camouflaged targets without prior location information. Providing such prompts essentially tells the model the target's location, which fundamentally contradicts the core challenge and assumption of camouflaged object detection. This design choice may render the experimental results unconvincing.

**Ethical Concerns:**

["NO or VERY MINOR ethics concerns only"]

**Final Justification:**

The authors have addressed all my concerns.

**Limitations:**

While I understand the authors propose prototypes to avoid occlusion and missing issues, I'm concerned whether the model might "overfit" to specific prototypes, thereby segmenting only parts of the target in subsequent frames rather than the complete object.

**Quality:**

3

**Strengths And Weaknesses:**

Strength:
1) The proposed architecture is well-illustrated with clear figures that effectively communicate the method's design.
2) The authors furnish a comprehensive comparison with state-of-the-art models, offering an intuitive overview.

Weakness:

Although this paper achieves good performance, it is based on box and 1-clip prompts, which may contradict the inherent nature of COD. Please refer to the questions below for more details.

---

> ### Author Rebuttal · Authors · 2025-07-31
>
> We sincerely thank you for the detailed and constructive feedback. We greatly appreciate your recognition of the clarity of our architectural design and the comprehensive comparisons provided. Now we address each of your concerns below.
>
> ---
>
> ## W1: Usage of prompts on VCOS
> Thank you for raising this important point. We would like to clarify that our work specifically addresses the task of VCOS, which has distinct goals from COD. COD typically focuses on detecting camouflaged objects in static images without any prior cues, whereas VCOS requires accurate and temporally consistent segmentation of camouflaged objects across entire video sequences. Although our method uses prompts, this design choice is aligned with the nature of SAM and SAM2 as promptable segmentation models. In fact, recent works such as COMPrompter[1], DSAM[2], and SAM-PM[3] also incorporate box or mask prompts to adapt SAM to the COD or VCOS task. Thus, using prompts is a common and well-accepted practice in recent work and does not contradict the objectives of the task.
>
> What's more, these two paradigms, COD for detection and VCOS for segmentation, can be effectively combined. For example, one could apply a COD model to the first frame of a video to generate a bounding box, which then serves as a box prompt to CamSAM2 for video segmentation. This integration not only makes CamSAM2 compatible with existing COD methods and fully automatic pipelines but also highlights the practical value and significance of our work in advancing VCOS. In real-world wildlife monitoring, search and rescue scenarios, a simple user interaction (e.g. clicking on the target object) can trigger CamSAM2 to segment and track it across time.
>
>
>
> ---
>
> ## Q1: Loss design clarification
> Thank you for this insightful question. While the original SAM2 parameters are frozen during training, we still apply supervision to its output $\mathbf{R}_i$ for the following reasons: Although SAM2's weights are fixed, the introduction of our learnable decamouflaged token changes the self-attention dynamics in the mask decoder. This token is concatenated with the original output tokens and jointly participates in self-attention and cross-attention layers (see Fig. 2(a) in the main paper). As a result, the outputs $\mathbf{R}_i$ are no longer identical to the original SAM2 outputs. Supervising them ensures that SAM2's outputs remain aligned with the segmentation intent, rather than being perturbed by the new token interactions. As shown in the Supple. Materials Table 4, we report the mIoU scores evaluated with the mask predicted by the SAM2 mask token and the decamouflaged token, the results also demonstrate the effectiveness of our design.
>
> ---
>
> ## Q2: Usage of prompts on VCOS
> Thanks for your question. Please refer to W1 in detail.
>
> ---
>
> ## Limitation: On prototype potential overfitting
> Thank you for raising this insightful concern. Our OPG and EOF modules are carefully designed to mitigate this risk. First, prototypes are dynamically generated from the predicted camouflaged regions using FPS and 1-iter $k$-means. This ensures that the prototypes are abstracted from the entire object masks, and are continually updated across frames and adapt to object appearance changes. Second, the EOF module uses a cross-attention mechanism, where the current frame's features query the prototype memory. This structure naturally suppresses outdated or irrelevant prototypes: if a prototype is inconsistent or suboptimal (e.g., due to occlusion or missing), it receives lower attention and contributes minimally to the output. This mechanism provides robustness and avoids overfitting to specific regions. Our Fig. 3 in the main paper further demonstrates this, showing that the segmentation results remain stable across time and do not overfit to a specific local region.
>
> ---
>
> References
> [1] COMPrompter: Reconceptualized Segment Anything Model with Multiprompt Network for Camouflaged Object Detection
> [2] Exploring Deeper! Segment Anything Model with Depth Perception for Camouflaged Object Detection
> [3] SAM-PM: Enhancing Video Camouflaged Object Detection using Spatio-Temporal Attention

---

> > ### Author Response · Authors · 2025-08-04
> >
> > Dear Reviewer,
> >
> > Thank you very much for your thoughtful feedback on our submission. We have carefully addressed your comments in our response.
> >
> > If you have any additional questions or concerns, we would be more than happy to clarify.
> >
> > We sincerely appreciate your time and efforts in reviewing our work.

---

> > ### Comment · Reviewer_rr2s · 2025-08-06
> > **Concerns Solved**
> >
> > The authors have mainly resolved my concerns, so I am willing to raise my score to Accept.

---

> > > ### Author Response · Authors · 2025-08-06
> > >
> > > Dear Reviewer,
> > >
> > > We are glad to hear that our response has resolved your concerns. Your comments were very helpful in improving the clarity and quality of our paper. We will include our response in the final version of the paper. We deeply appreciate your time and effort!

---

### Official Review · Reviewer_TQ6w · 2025-07-01

**Clarity:** 3
**Significance:** 3
**Originality:** 3
**Rating:** 5
**Confidence:** 4

**Summary:**

This paper proposes a method to adapt SAM2 for video camouflaged object segmentation by using a learnable de-camouflaged token to optimize features for this task without modifying the base architecture's parameters. The method also utilizes high-resolution features from the visual encoder to capture fine details. Additionally, it leverages high-resolution information from previous frames through a newly introduced object prototype generation module. Empirical validation is conducted on three datasets to support the proposed claims.

**Questions:**

Some suggestions:

* To better highlight the novelty of the contribution, the major components (e.g., Decam. tokens, OPG) and minor components (e.g., IOF, EOF) could be clearly distinguished. Clarifying this distinction would help readers appreciate the significance of the major contributions without having them overshadowed by the minor ones.

* The rationale behind certain design choices could be further elaborated. For instance, was the use of 1×256 Decam. tokens determined through ablation studies or based on intuition? Similarly, was the value of k=5 tuned using MoCA-Mask and then applied uniformly across all datasets?

**Ethical Concerns:**

["NO or VERY MINOR ethics concerns only"]

**Final Justification:**

The authors' response has adequately addressed the concerns raised in my initial review. Accordingly, I am pleased to revise my rating in recognition of their efforts. I extend my best wishes for the success of their work.

**Limitations:**

Limitations are in supplementary document.

**Quality:**

3

**Strengths And Weaknesses:**

Strengths:

* The paper is well-presented, with a clear and logical flow that makes it easy to follow. Figure 1 effectively illustrates the main architectural modifications at a high level, providing an intuitive understanding of the proposed approach. Figure 2 clearly explains the detailed architecture. Overall, the writing is coherent and contributes to the paper’s presentation.

* The paper provided mathematical formulations with appropriate notations to explain the proposed components.

* The paper supported the claims with comprehensive experiments on three benchmark datasets and more ablation experiments. The results showed noticeable improvement over SOTA approaches and SAM2.


Weaknesses:

* Based on Figure 2 and Equation 1, the proposed IOF module appears to be a customization of the Feature Pyramid Network, which is widely used in segmentation models as an additional component to generate high-resolution features. I might be missing something, but I could not identify the novelty of this component.

* Certain design choices that could be important for reproducing or extending the work are not explained or justified in the main text—such as the shape of the Decam. tokens, whether the number of prototypes is fixed or adaptive across datasets, and how these values are determined.

* The comparison between the CamSAM2 mask, the SAM2 mask, and their average is crucial for understanding the specific improvements or capabilities introduced by CamSAM2. However, this analysis is missing from the main text and is only presented in the supplementary material.

* Minor typos, e.g., line 109: promotable -> promptable.

---

> ### Author Rebuttal · Authors · 2025-07-31
>
> We are glad to hear that our paper is well-presented, with clear architectural diagrams, coherent mathematical formulations, and strong empirical validation across multiple datasets. Below, we address your concerns point by point.
>
> ## W1: IOF novelty clarification
> We appreciate your comment. The IOF module is designed to address a key limitation in SAM2, where the early-stage high-resolution features are not fully utilized during the segmentation process. These features contain fine-grained texture and boundary details are especially crucial for VCOS. To exploit this, IOF fuses early-layer high-resolution features with memory-conditioned semantic features, thereby enhancing the model's ability to perceive subtle camouflage cues. While IOF may share structural similarities with general fusion mechanisms (e.g., FPN), its motivation, integration strategy, and application context are specifically tailored for VCOS. To the best of our knowledge, this is the first work to incorporate early-layer high-resolution features into SAM2 for VCOS. Our ablation study (Table 5 in the main paper) shows that adding IOF brings a non-trivial performance gain (e.g., +0.5 mIoU), highlighting both its necessity and effectiveness.
>
> ---
>
> ## W2: Explanation of design choices
> Thanks for pointing this out. We clarify these design decisions as follows:
>
> 1. Decamouflaged token: The decamouflaged token is designed with a shape of 1 × 256. As illustrated in Fig. 2 in the main paper, this decamouflaged token extends SAM2's token structure, aligning with the default embedding dimension of output tokens in SAM2. This ensures compatibility with the existing self-attention structure without modifying SAM2's architecture.
> 2. Number of prototypes in OPG: The number of prototypes in OPG is fixed (set to k = 5) across all datasets. We initially tuned this value on MoCA-Mask and found it to generalize well, as shown in our ablation study in the main paper Table 7. Our goal is not precise clustering, but to abstract object-aware prototypes to guide cross-attention in EOF. A fixed k also simplifies inference and avoids introducing dataset-specific hyperparameters.
>
> We will clearly add these explanations in the final version. In addition, we will publicly release our code to support reproducibility and transparency.
>
> ---
>
> ## W3: On the placement of mask comparisons
> Thanks for the suggestion. In the final version, we will consider moving this analysis into the main paper to better showcase how our design benefits segmentation accuracy.
>
> ---
>
> ## W4: Minor typo
> Thanks for pointing out our typo. We will fix it and carefully proofread the entire manuscript to eliminate other minor errors in the final version.
>
> ---
>
> ## Q1: Some suggestions
> Thank you for the suggestion. We agree that distinguishing major and minor components can help clarify our contributions. We will emphasize this distinction more clearly in the revised version.
>
> ---
>
> ## Q2: Explanation of design choices
> Thanks for your question. Please refer to W2 in detail.

---

> > ### Author Response · Authors · 2025-08-04
> >
> > Dear Reviewer,
> >
> > Thank you very much for your thoughtful feedback on our submission. We have carefully addressed your comments in our response.
> >
> > If you have any additional questions or concerns, we would be more than happy to clarify.
> >
> > We sincerely appreciate your time and efforts in reviewing our work.

---

### Official Review · Reviewer_o2wz · 2025-07-20

**Clarity:** 3
**Significance:** 2
**Originality:** 2
**Rating:** 2
**Confidence:** 5

**Summary:**

This paper proposes the CamSAM2 model, which enhances SAM2 for video camouflaged object segmentation (VCOS) by introducing several key components: a learnable de-camouflaging token, an Implicit Object-aware Fusion (IOF) module, an Explicit Object-aware Fusion (EOF) module, and an Object Prototype Generation (OPG) module. The model retains all original SAM2 parameters and introduces only a small number of learnable parameters. Despite this, CamSAM2 achieves strong performance on multiple datasets (MoCA-Mask, CAD, SUN-SEG), while preserving SAM2’s zero-shot generalization and natural video segmentation capabilities.

**Questions:**

Please see **Weaknesses**.

**Ethical Concerns:**

["NO or VERY MINOR ethics concerns only"]

**Final Justification:**

Thank you to the authors for the detailed and thoughtful rebuttal. I appreciate the effort and clarity with which the response was written.

However, after carefully considering the rebuttal and revisiting the paper, I find that several of my core concerns remain unaddressed. I elaborate on them below:

**Lack of Conceptual Novelty**

The core components of CamSAM2—namely the decamouflaged token, Implicit/Explicit Object-aware Fusion (IOF/EOF), and Object Prototype Generation (OPG)—are essentially adaptations or recombinations of widely established techniques, commonly used in segmentation and video object tracking:

- **IOF**, which fuses multi-scale features, follows a standard design pattern found in numerous architectures such as *UPerNet* and *CamoFormer* (TPAMI 2024), both of which target camouflaged object detection.
- The proposed **decamouflaged token** functions similarly to **virtual tokens** or **learnable prompts**, as explored in recent works such as *SAM-PM* [1] and *SAM-Adapter* [2]. These prompt-based fine-tuning strategies are now commonplace in SAM-derived models.
- The **OPG** module, based on *Farthest Point Sampling* (FPS) and *k-means clustering*, is highly reminiscent of prototype-based memory techniques seen in models like *AOT* [3] and *XMem* [4], which have been extensively studied in video object segmentation and memory-augmented frameworks.

In essence, the method does not introduce novel algorithmic principles or modeling paradigms. Rather, it assembles previously known components into a task-specific pipeline. While this engineering effort may be useful in practice, it does not meet the NeurIPS bar for conceptual or methodological innovation.

References

 [1] *SAM-PM: Enhancing Video Camouflaged Object Detection using Spatio-Temporal Attention*, CVPRW 2024

 [2] *SAM-Adapter: Adapting SAM in Underperformed Scenes*, arXiv 2023

 [3] *AOT: Associating Objects with Transformers for Video Object Segmentation*, NeurIPS 2021

 [4] *XMem: Long-Term Video Object Segmentation with an Atkinson-Shiffrin Memory Model*, ECCV 2022



**Misleading Claim of Being “First SAM2 for VCOS”**

The submission claims to be the *first* to adapt SAM2 for video camouflaged object segmentation (VCOS). However, this is contradicted by existing literature:

- [5] *When SAM2 Meets Video Camouflaged Object Segmentation*, which presents a comprehensive evaluation and adaptation of SAM2 for VCOS.
- [6] Tang & Li, *Evaluating SAM2’s Role in Camouflaged Object Detection: From SAM to SAM2*, which also proposes SAM2 adaptations for COD and VCOS.

Both works predate CamSAM2 and are cited in the submission. The “first” claim is therefore factually inaccurate and weakens the originality and positioning of the paper.



**Trivial Task-Specific Design**

The proposed modifications are narrowly tailored for VCOS and show limited signs of generality:

- There is no evidence that the approach extends to other fine-grained segmentation tasks or that the components (e.g., decamouflaged token, OPG) offer improvements beyond the VCOS setting.

Thus, the framework appears to be an ad hoc design for a specific downstream task, assembled from task-specific heuristics and lacking broader impact or general-purpose applicability.



Despite solid empirical performance on VCOS benchmarks, the paper’s contributions remain derivative, task-specific, and lacking in conceptual depth. The empirical results alone do not compensate for the absence of methodological novelty or general insight.  Accordingly, I have decided to lower my score.

**Limitations:**

yes

**Paper Formatting Concerns:**

None.

**Quality:**

2

**Strengths And Weaknesses:**

**Strengths**

- Strong empirical results: Demonstrated superior performance compared to SAM2 on several standard benchmarks.
- Good generalization: Evaluation on multiple datasets shows consistent improvements and robustness across diverse scenarios.

---

**Major Weaknesses**

1. Limited novelty: The main contributions are combinations of existing techniques:
- IOF performs cross-layer feature fusion, a well-known strategy.
- The de-camouflaging token is similar to virtual tokens used in fine-tuning.
- EOF employs explicit prototype refinement, while OPG relies on standard clustering techniques (e.g., farthest point sampling, k-means).

Overall, the proposed components are adaptations or recombinations of established ideas, limiting the conceptual novelty.

2. Potential inefficiency: The introduction of multiple modules, especially OPG with FPS and k-means clustering, may introduce considerable inference overhead, raising concerns about deployment in time-sensitive applications.

3. Insufficient analysis and ablations:

- No exploration of different numbers of de-camouflaging tokens.
- Lack of analysis on the impact of different sampling and clustering strategies.
- No prototype visualization or interpretability discussion, which would help understand the model's internal behavior.

---

**Minor Weaknesses**

- Figure 2: The component labeled "I" is undefined, and the de-camouflaging token visualization is confusing.

- Unsupported claims: The model is claimed to retain natural image segmentation performance, but no experimental results are provided to validate this.

- Figure 3: Ground truth (GT) visualizations contain potential errors or unclear rendering that could mislead interpretation.

---

> ### Author Rebuttal · Authors · 2025-07-31
>
> Thank you for your valuable suggestions and for recognizing the strengths of our work, including the strong empirical performance and the generalization ability of CamSAM2 across diverse VCOS scenarios. Below, we respond in detail to your concerns.
>
> ---
>
> ## W1. Novelty clarification
> CamSAM2 introduces a cohesive and task-specific redesign tailored for the challenging VCOS task. Concretely, CamSAM2 addresses two key limitations of SAM2 in the VCOS scenarios: (i) SAM2 is optimized for natural scene segmentation and struggles with the fine-grained, low-contrast textures characteristic of camouflaged objects; (ii) SAM2's memory module stores only coarse, low-resolution features, which impairs its ability to maintain temporal consistency across time.
>
> To tackle these, we introduce a set of interdependent modules, which are co-designed rather than independently stacked. This integration in a frozen SAM2 pipeline is novel and non-trivial. To the best of our knowledge, CamSAM2 is the first work to successfully and efficiently adapt SAM2 to the VCOS task, achieving both architectural innovation and practical performance gains.
>
> ---
>
> ## W2. Efficiency concerns of OPG (FPS and k-means)
> Thank you for highlighting the potential runtime overhead introduced by our proposed modules, especially the OPG. To address this, we profile CamSAM2 with hiera-T as the backbone on a 106-frame camouflaged video from the MoCA-Mask test set, and report the average per-frame latency in Table 1. For comparison, we also include the original SAM2 per-frame runtime.
>
> | Component   | Avg Time (ms) | Relative Percentage |
> |:-------------|:---------------:|:---------------------:|
> | SAM2        | 83.7          | 100.0%              |
> | FPS     | 3.5           | 4.2%                |
> | k-means | 1.0           | 1.2%                |
> | OPG         | 4.6           | 5.5%                |
> | CamSAM2     | 89.7          | 107.2%              |
>
> **Table 1. Average runtime per frame and overhead breakdown of CamSAM2 compared to original SAM2.**
>
> These results confirm that CamSAM2 introduces only 6 ms (7.2%) additional latency per frame compared to SAM2. As you correctly noted, the OPG module (comprising both FPS and 1-iter *k*-means clustering) is indeed the dominant component of overhead. Apart from OPG, all other components, including IOF, EOF, and associated processing, together contribute only approximately 1.4 ms (1.7%) on average per frame. These results demonstrate that CamSAM2 remains efficient and is suitable for practical deployment, even in time-sensitive VCOS applications.
>
> ---
>
> ## W3. Insufficient analysis and ablations
>
> ### 1. The number of the decamouflaged token
> Thanks for your suggestion. We conduct an ablation study by increasing the number of tokens to 3 and observed a comparable result in mIoU, as shown in Table 2. Adding more decamouflaged tokens does not structurally improve the representation quality but may introduce redundancy and computational cost.
>
>
> | Model    | Number of Decam. Token | mIoU |
> |:-------------|:---------------:|:---------------------:|
> | SAM2     | -                       | 44.8 |
> | CamSAM2  | 3                       | 53.3 |
> | CamSAM2  | 1                       | **54.6** |
>
> **Table 2. Impact of using different numbers of decamouflaged tokens.**
>
> ---
>
> ### 2. Sampling and clustering strategies
> To assess the robustness and generality of OPG, we experiment on different sampling and clustering strategies:
> (i) Farthest Point Sampling (FPS) vs. Average Sampling for selecting initial cluster centers, and (ii) *k*-means vs. Gaussian Mixture Model (GMM) as the clustering algorithm.
>
> As shown in Table 3, all combinations yield reasonable results. Compared to average sampling, FPS better preserves spatial diversity among selected features, leading to more representative and discriminative prototypes. These results still confirm that OPG is flexible and not overly sensitive to specific implementation choices, which enhances its applicability.
>
>
> | Sampling         | Clustering  | mIoU |
> |:-------------|:---------------:|:---------------------:|
> | Average Sampling | *k*-means   | 52.1 |
> | Average Sampling | GMM         | 52.7 |
> | FPS              | GMM         | 53.6 |
> | **FPS**          | ***k*-means** | **54.6** |
>
> **Table 3. Impact of different sampling and clustering strategies in OPG.**
>
> ---
>
> ### 3. Prototype visualization and discussion
> Thanks for pointing this out. Following your suggestion, we compute the similarity map between each prototype of previous frames and the feature map of the current frame. These maps reflect how each prototype activates spatially over the image.
>
> Our visualizations show that different prototypes consistently activate on semantically meaningful object regions. For example, the prototype shows strong activations (high similarity) precisely on the camouflaged object, while suppressing the background. This suggests that the prototypes successfully abstract object-aware semantics, supporting their effectiveness in guiding the cross-attention in EOF. Although visual examples cannot be included in this rebuttal, we will add these visualizations and related analysis in the final version of the paper.
>
> ---
>
> ## Minor Weaknesses
>
> 1. Thanks for your suggestions, we will refine the figures in the final version.
>
> 2. We appreciate your attention to this point. To clarify, in Sec. 7 of the Supple. Material, we describe the mode switch between CamSAM2 and SAM2 at inference. When the toggle is off, all parameters introduced by CamSAM2 are deactivated, and the model functions identically to SAM2 without any parameter modifications. We also test on the SA-V test set and get the same J\&F result. This design ensures full architectural compatibility and serves as the basis for our claim about retaining the original segmentation capabilities on natural images and videos.
>
> 3. Thank you for pointing this out. The camouflaged objects in our benchmark are indeed highly challenging to perceive due to their nature. The ground-truth masks in Fig. 3 in the main paper are sourced directly from the MoCA-Mask dataset, which provides high-quality human-annotated segmentation masks. While the visualization may appear unclear in certain cases because the objects are highly camouflaged, the annotations are accurate and widely accepted in the community.

---

> > ### Comment · Reviewer_o2wz · 2025-08-07
> >
> > Thank you to the authors for the detailed and thoughtful rebuttal. I appreciate the effort and clarity with which the response was written.
> >
> > However, after carefully considering the rebuttal and revisiting the paper, I find that several of my core concerns remain unaddressed. I elaborate on them below:
> >
> > **Lack of Conceptual Novelty**
> >
> > The core components of CamSAM2—namely the decamouflaged token, Implicit/Explicit Object-aware Fusion (IOF/EOF), and Object Prototype Generation (OPG)—are essentially adaptations or recombinations of widely established techniques, commonly used in segmentation and video object tracking:
> >
> > - **IOF**, which fuses multi-scale features, follows a standard design pattern found in numerous architectures such as *UPerNet* and *CamoFormer* (TPAMI 2024), both of which target camouflaged object detection.
> > - The proposed **decamouflaged token** functions similarly to **virtual tokens** or **learnable prompts**, as explored in recent works such as *SAM-PM* [1] and *SAM-Adapter* [2]. These prompt-based fine-tuning strategies are now commonplace in SAM-derived models.
> > - The **OPG** module, based on *Farthest Point Sampling* (FPS) and *k-means clustering*, is highly reminiscent of prototype-based memory techniques seen in models like *AOT* [3] and *XMem* [4], which have been extensively studied in video object segmentation and memory-augmented frameworks.
> >
> > In essence, the method does not introduce novel algorithmic principles or modeling paradigms. Rather, it assembles previously known components into a task-specific pipeline. While this engineering effort may be useful in practice, it does not meet the NeurIPS bar for conceptual or methodological innovation.
> >
> > References
> >
> >  [1] *SAM-PM: Enhancing Video Camouflaged Object Detection using Spatio-Temporal Attention*, CVPRW 2024
> >
> >  [2] *SAM-Adapter: Adapting SAM in Underperformed Scenes*, arXiv 2023
> >
> >  [3] *AOT: Associating Objects with Transformers for Video Object Segmentation*, NeurIPS 2021
> >
> >  [4] *XMem: Long-Term Video Object Segmentation with an Atkinson-Shiffrin Memory Model*, ECCV 2022
> >
> >
> >
> > **Misleading Claim of Being “First SAM2 for VCOS”**
> >
> > The submission claims to be the *first* to adapt SAM2 for video camouflaged object segmentation (VCOS). However, this is contradicted by existing literature:
> >
> > - [5] *When SAM2 Meets Video Camouflaged Object Segmentation*, which presents a comprehensive evaluation and adaptation of SAM2 for VCOS.
> > - [6] Tang & Li, *Evaluating SAM2’s Role in Camouflaged Object Detection: From SAM to SAM2*, which also proposes SAM2 adaptations for COD and VCOS.
> >
> > Both works predate CamSAM2 and are cited in the submission. The “first” claim is therefore factually inaccurate and weakens the originality and positioning of the paper.
> >
> >
> >
> > **Trivial Task-Specific Design**
> >
> > The proposed modifications are narrowly tailored for VCOS and show limited signs of generality:
> >
> > - There is no evidence that the approach extends to other fine-grained segmentation tasks or that the components (e.g., decamouflaged token, OPG) offer improvements beyond the VCOS setting.
> >
> > Thus, the framework appears to be an ad hoc design for a specific downstream task, assembled from task-specific heuristics and lacking broader impact or general-purpose applicability.
> >
> >
> >
> > Despite solid empirical performance on VCOS benchmarks, the paper’s contributions remain derivative, task-specific, and lacking in conceptual depth. The empirical results alone do not compensate for the absence of methodological novelty or general insight.  Accordingly, I have decided to lower my score.

---

> > > ### Author Response · Authors · 2025-08-08
> > >
> > > Dear Reviewer,
> > >
> > > We regret to hear that while your comments do not raise new fatal concerns and you appreciate our detailed and thoughtful rebuttal, you still decided to reduce the score.
> > >
> > > We respectfully do not agree with the points you raised in the rebuttal and provide the response below.
> > >
> > > 1. Regarding "Lack of Conceptual Novelty"
> > >
> > >    It is worth noting that we do not claim conceptual novelty of the proposed modules in this paper, rather, the novelty of CamSAM2 lies in a unified and task-specific redesign of SAM2 for the demanding and important VCOS task while inheriting all the abilities of the original SAM2. The proposed co-designed and interdependent modules are organically integrated into a frozen SAM2 pipeline, effectively adapting the famous foundation model for VCOS, which is validated by our significant performance gains over the original SAM2 and SAM2-FT (finetuning of SAM2). The switch between performing VCOS and general video segmentation can be easily done as shown in Fig.1 in the supple. material. Therefore, our method is practically useful for different kinds of VCOS applications such as wildlife monitoring and search-and-rescue scenarios as well as general video segmentation.
> > >
> > >
> > > 2. Regarding "Misleading Claim of Being 'First SAM2 for VCOS'"
> > >
> > >    We respectfully do not agree with this point.
> > >
> > >    First, when this paper was submitted, both the works you mentioned ([5] and [6]) had not been formally published. Second, our original claim is "CamSAM2 is the first work to successfully and efficiently adapt SAM2 to the VCOS". In the context, what we mean by adapting SAM2 is adding new modules (trainable parameters) and conducting sophisticated optimization, rather than simple finetuning like [5] and simple evaluation like [6]. Third, our original claim is in this rebuttal, rather in the main submission.
> > >
> > > 3. Regarding "Trivial Task-Specific Design"
> > >
> > >    We respectfully do not agree with your point "the proposed modifications are narrowly tailored for VCOS and show limited signs of generality".
> > >
> > >    It is true that our method is tailored for VCOS. However, CamSAM2 inherits all of SAM2's abilities. Performing a simple switch can result in the predictions of the original SAM2. That's to say, CamSAM2 extends SAM2's ability on the important task of VCOS without losing any generality.

---

> > > > ### Comment · Reviewer_o2wz · 2025-08-08
> > > >
> > > > Thank you again for the rebuttal. I appreciate the authors' efforts in addressing the concerns. However, after reading the revised response carefully, I find that my core issues regarding novelty, positioning, and generality remain insufficiently addressed. I elaborate on each point below.
> > > >
> > > >
> > > > 1. Lack of Novelty
> > > >
> > > > Thank you for the clarification. However, I remain unconvinced by the response.
> > > >
> > > > While the authors explicitly acknowledge that their work does not claim conceptual novelty, their justification—namely, a “unified and task-specific redesign” of SAM2—does not adequately address the core concern. As highlighted in my initial review, each of the proposed components (IOF, EOF, decamouflaged token, OPG) is based on well-established techniques. The integration of these elements into a SAM2 pipeline, even if task-aligned, does not constitute a methodological contribution on its own.
> > > >
> > > > Simply adapting or recombining existing modules for a specific task (VCOS in this case) does not meet the threshold of innovation expected at NeurIPS. Many prior works have already adapted foundation models like SAM using prompt tuning, memory mechanisms, and multi-scale fusion. Without a novel algorithmic insight, theoretical advancement, or a fundamentally new architectural component, the work reads more like an incremental engineering effort than a research contribution.
> > > >
> > > > Performance gains alone—while practically valuable—are not sufficient to justify publication in a top-tier venue unless they stem from a new and generalizable idea. The rebuttal does not present such an idea, and therefore, my concern about the lack of conceptual novelty remains unresolved.
> > > >
> > > >
> > > >
> > > >
> > > >
> > > > 2. Misleading Claim of Being “First SAM2 for VCOS”
> > > >
> > > > I respectfully disagree with the authors’ rebuttal on this point.
> > > >
> > > > Although [5] (*When SAM2 Meets Video Camouflaged Object Segmentation*) and [6] (*Tang & Li, Evaluating SAM2’s Role in Camouflaged Object Detection*) were not formally published, both have been publicly available on arXiv for nearly a year. As early efforts exploring the intersection of SAM2 and VCOS, these works should have been properly cited, discussed, and compared against.
> > > >
> > > > Importantly, [5] is not merely a “simple finetuning” study. It presents:
> > > >
> > > > - A unified adaptation pipeline of SAM2 under multiple fine-tuning modes (Automatic, Semi-supervised, MLLM + SAM2, and VCOS + SAM2),
> > > > - Strong empirical results (e.g., 77.5 on CAD), which surpass both the original SAM2 (66.7) and CamSAM2 (69.2),
> > > >
> > > > Furthermore, the rebuttal is also part of your work and should not be separated from the main paper.

---

> > > > > ### Author Response · Authors · 2025-08-09
> > > > >
> > > > > Dear Reviewer,
> > > > >
> > > > > Thank you again for your reply! We respectfully do not agree with the points you raised in the rebuttal and provide the response below.
> > > > >
> > > > >  1. Response to the claim regarding "Lack of Novelty"
> > > > >
> > > > > We respectfully disagree with your opinion that the integration of our modules does not constitute a methodological contribution. In methodological research, novelty does not require every component to be invented from scratch; it can arise from a new architectural paradigm that systematically addresses limitations of existing approaches. As clarified in our previous response, the novelty of CamSAM2 lies in a unified, task-specific redesign of SAM2 for the demanding VCOS task, while retaining all capabilities of the original SAM2. The co-designed, interdependent modules are embedded directly into SAM2’s internal processing flow, extending its capability to handle VCOS-specific challenges.
> > > > >
> > > > > Moreover, this architectural design is generalizable. As demonstrated on camouflaged animal videos (MoCA-Mask) and polyp videos (SUN-SEG), it consistently improves segmentation performance across distinct domains that share low-contrast, fine-grained characteristics. This indicates that the performance gains are not the sole contribution but are a direct result of our proposed redesign rather than an incremental engineering effort. Although our methodological contribution was originally tailored for VCOS, it can also be applied to other video segmentation tasks with similar challenges.
> > > > >
> > > > > 2. Response to the claim regarding "First SAM2 for VCOS" and [5]:
> > > > >
> > > > > We thank you for the follow-up, but we respectfully disagree with the statement that our rebuttal was misleading. Our position remains that [5] is essentially a simple fine-tuning study, and we have already cited it in the appropriate place in our submission. We also fine-tuned SAM2 for direct and fair comparison, as we already mentioned in lines 235-236 and Table 1 in the main paper.
> > > > >
> > > > > What's more, there are two factual inaccuracies in your comment:
> > > > >
> > > > > (i) Nature of adaptation in [5]
> > > > >
> > > > > As we mentioned in the last message, adapting SAM2 means introducing new trainable modules into the architecture and conducting sophisticated optimization. In contrast, [5] does not perform fine-tuning in any of the four modes as you mentioned (Automatic, Semi-supervised, MLLM+SAM2, VCOS+SAM2). Specifically:
> > > > >
> > > > > - The Automatic and Semi-supervised results in [5] are evaluations of SAM2 directly without weight updates.
> > > > >
> > > > > - The MLLM+SAM2 and VCOS+SAM2 modes in [5] simply use MLLM or other VCOS model outputs as prompts; these approaches involve no SAM2 internal architectural changes or weight updates.
> > > > >
> > > > > - The only fine-tuning in [5] is direct fine-tuning on VCOS datasets, which we have done and reported in Table 1.
> > > > >
> > > > > (ii) Misinterpretation of results
> > > > >
> > > > > The "77.5 on CAD" reported in [5] is achieved by evaluating the SAM2-Large model using the mask prompt. In our Table 2, results are reported using the SAM2-Tiny model and different prompt types. Directly comparing these numbers conflates different model scales and prompt settings. Moreover, as noted in Section 3.1 of [5], its CAD results were produced after removing some incorrect annotations from the dataset. In our evaluation, we did not remove any annotations. Both evaluation settings are used in recent literature and are accepted by the community. Importantly, when we run the released code of [5] without removing incorrect annotations, the results match those we reported, which is reasonable because this setting corresponds to the original SAM2 evaluation.
> > > > >
> > > > > In summary, while [5] explores SAM2 in VCOS, its methodology and scope differ substantially from ours. Our claim of novelty refers specifically to introducing new modules and tailored optimization for VCOS, rather than merely combining existing models without modifying SAM2’s architecture or simply fine-tuning its weights.

---

> > > > > > ### Comment · Reviewer_o2wz · 2025-08-09
> > > > > >
> > > > > > Thank you for the additional clarifications. However, my core concerns remain unresolved.
> > > > > >
> > > > > > In essence, this work suffers from a **fundamental lack of novelty**. All key components (IOF, EOF, decamouflaged token, OPG) are direct adaptations of well-established designs—UPerNet, CamoFormer, SAM-Adapter, AOT, XMem—several of which (e.g., UPerNet, CamoFormer) **were originally developed for camouflaged object recognition**. The integration of these existing techniques into SAM2 is technically straightforward and does not introduce any new algorithmic principles or architectural innovations.
> > > > > >
> > > > > > In light of the above, I consider this work does not meet the bar of NeurIPS. My recommendation remains Reject.

---

> ### Author Response · Authors · 2025-08-04
>
> Dear Reviewer,
>
> Thank you very much for your thoughtful feedback on our submission. We have carefully addressed your comments in our response.
>
> If you have any additional questions or concerns, we would be more than happy to clarify.
>
> We sincerely appreciate your time and efforts in reviewing our work.

---

### Note · Authors · 2025-08-12

Dear AC,

We sincerely thank you and all reviewers for their time, effort, and constructive feedback throughout the review process.

We greatly appreciate the recognition from three reviewers regarding the strengths of our work. Two reviewers (TQ6w and rr2s) gave positive ratings, and Reviewer L184 expressed the intention to increase their rating after our rebuttal, acknowledging that our rebuttal had fully addressed their concerns. We regret that Reviewer o2wz kept a negative rating, but we respectfully disagree and clarify:

(1) Our approach is distinct from the methods referenced by the reviewer. Specifically:
- Decam token: While the reviewer noted that decam token appears similar to learnable prompts, its design and integration differ from SAM2-Adapter. SAM2-Adapter injects prompts into **every encoder block**, but we introduce **a single token** only in the mask decoder, extending SAM2's token structure for the camouflage perception.
- OPG: It is built on SAM2's memory mechanism: rather than storing multiple types of memories as in XMem or maintaining long/short-term attentions over identification embeddings as in AOT, we retain only a few compact prototypes per object and match them once via cross-attention in EOF. This lightweight approach reduces complexity while boosting SAM2's perception of camouflaged objects.
- IOF: Unlike UPerNet, which concatenates multi-scale features before decoding, and CamoFormer, which runs a multi-stage decoder with masked separable attention and multiple outputs, IOF applies a lightweight compression to multi-scale features and fuses them once with the embedding. This injects fine, high-res details for VCOS while keeping SAM2 frozen, avoiding repeated decoding and improving efficiency.

(2) The task we address is different. Methods like SAM2-Adapter and CamoFormer focus on **image-based** COD, whereas our work tackles **video** camouflaged object segmentation, which presents additional temporal and consistency challenges.

(3) CamSAM2 demonstrates broad applicability. It achieves substantial gains in both natural-scene camouflage (up to 12.2 mDice on MoCA-Mask) and medical camouflage (up to 19.6 mDice on SUN-SEG-Hard), highlighting its effectiveness across diverse domains.

Given strong reviewer support, large empirical gains, and our methodological clarifications, we believe CamSAM2 makes a practically useful and methodologically thoughtful contribution to adapting SAM2 for VCOS.

Thank you for your consideration.

---

### Decision · Program_Chairs · 2025-09-17

**Decision:**

Accept (poster)

**Comment:**

This paper proposes **CamSAM2**, an adaptation of SAM2 for camouflaged object segmentation in videos.

The submission initially received **2 × Borderline Accept** and **2 × Borderline Reject** ratings. During the rebuttal period, the authors and reviewers engaged actively, with the authors providing additional analyses and ablations. After the discussion, **three reviewers raised their scores to Accept**, while **one reviewer lowered their score to Reject**, citing concerns around novelty and positioning.

While the AC acknowledges these concerns, the overall framework and integration underlying CamSAM2 are considered **meritorious**. Given its strong performance gains and practical impact, the AC finds the paper to be an **impactful contribution** and therefore recommends **Accept**. Congratulations.

The authors are encouraged to **revise the framing in the final version** by:
1. Clearly positioning CamSAM2 with respect to prior work on camouflaged object segmentation.
2. Incorporating the additional results and discussions provided during the rebuttal period to further strengthen the paper.